# The genetic and biochemical basis of human leading strand synthesis

Alessandro Agnarelli [1,6], Lauryn Buckley-Benbow [1,6], Meryem Ozgencil [1], Melanie Lad[2], Khamal Kwesi Ampah[2], Alex Kalinka[2], Ondrej Belan[3], Sarah Maslen[4], Mark J. Skehel[4], David Walter [2], Matthew Day [5] ✉ & Roberto Bellelli [1] ✉

The maintenance of genome stability requires efficient leading strand synthesis by DNA Polymerase Epsilon (Polε). By performing CRISPR genetic screens in cells lacking the POLE4 subunit of Polε we define a genetic map of the factors required to support Polε function in the absence of its accessory subunits. A set of genes involved in iron metabolism emerge as required to sustain Iron Sulphur Cluster (ISC)-dependent Polε activity. We then dissect a synthetic lethal interaction between POLE3-POLE4 and the CHTF18-RFC2/5 complex. By combining cell biology, structural modelling and biochemistry, we define the existence of two tiers of regulation of Polε processivity: leading strand-specific loading of PCNA by CHTF18-RFC2/5 and "gripping" of newly synthesised dsDNA by POLE3-POLE4. The combined loss of these functions is incompatible with leading strand synthesis and viability. In summary, we describe the biochemical basis of human leading strand synthesis and the consequence of its dysfunction in genome stability.

The maintenance of genome stability requires accurate and efficient DNA replication and dysfunction of this process is associated with human genetic disease and cancer[1,2]. At the core of the replication machinery is DNA Polymerase Epsilon (Polε), a four subunit complex (POLE1-4), required for assembly of the CMG (CDC45/MCM2-7/GIN1-4) helicase and leading strand DNA synthesis[3]. How Polε carries out continuous and uninterrupted leading strand synthesis in human cells remains to be defined. Biochemical reconstitution with purified human proteins, has recently revealed a minimum set of factors required for efficient leading strand synthesis in vitro[4]. Importantly, the alternative clamp loader complex composed of CHTF18 (CHromosome Transmission Fidelity factor 18), CHTF8 (CHromosome Transmission Fidelity factor 8), DSCC1 (DNA Replication and Sister Chromatin Cohesion 1) and RFC2/5 (Replication Factor 2/5) (hereafter referred to as

CHTF18-RFC complex) emerged as crucially required to sustain in vitro human leading strand replication[4]. Despite this, if and how loss of the CHTF18-RFC complex impacts replication dynamic in human cells remain to be defined.

In addition to POLE1, its catalytic subunit, and POLE2, which structurally links Polε to the CMG, Polε harbours two "accessory" subunits, POLE3 and POLE4, which constitutively heterodimerize via histone fold motifs and support histone H3-H4 recycling during DNA replication[5–7]. We previously showed that loss of POLE3-POLE4 in mouse and human cells also leads to reduced levels of Polε, in association with signs of reduced origin activation in primary but not immortalized and cancer cells[8,9]. Importantly, cancer cells lacking POLE3-POLE4, despite being fully competent for origin activation, present a remarkable sensitivity to ATR and PARP inhibitors; more

[1]Centre for Cancer Cell & Molecular Biology, Barts Cancer Institute, Queen Mary University of London, Charterhouse Square, EC1M 6BQ, London, UK. [2]Joint Cancer Research Horizons-AstraZeneca Functional Genomics Centre, Biomedical Campus, 1 Francis Crick Avenue, CB2 0AA Cambridge, UK. [3]Division of Genetics, Department of Medicine, Brigham and Women's Hospital and Department of Genetics, Harvard Medical School, Boston, MA 02115, USA. [4]Proteomics Science Technology Platform, The Francis Crick Institute, 1 Midland Road, NW1 1AT London, UK. [5]Centre for Molecular Cell Biology, School of Biological and Behavioural Sciences, Blizard Institute, Queen Mary University of London, London E1 2AT, UK. [6]These authors contributed equally: Alessandro Agnarelli, Lauryn Buckley-Benbow. ✉e-mail: matthew.day@qmul.ac.uk; r.bellelli@qmul.ac.uk

specifically, PARPi treatment in POLE3-POLE4 KO cells unleashes replicative gap accumulation and leads to BRCA1-independent sensitization to PARPi[10,11]; these data are independent of Polε expression levels thus suggesting additional functions for POLE3-POLE4 in controlling DNA replication and genome stability. The nature of these functions remain to be defined.

Despite inclusion of the full Polε complex in cryoEM studies in the presence of the CMG[12,13], or the sliding DNA clamp PCNA[14,15], a structure of full human Polε is currently missing, likely due to the inherently high flexibility of the complex. While this evidence points to a rather flexible structure of Polε holoenzyme, work in budding yeast has suggested the existence of a more rigid conformation of Polε supported by Dpb3-Dpb4, the yeast orthologs of POLE3-POLE4, acting as a wedge sitting between the catalytic and non catalytic modules of Pol2, the catalytic subunit of budding yeast Polε[16]. While it was initially proposed this might represent a processive conformation of the polymerase, it is worth noting that this conformation was not observed in a yeast CMG-E structure[12], nor is it compatible with PCNA binding[14]; as such, it is unlikely it might exist in the context of established replication forks. Furthermore, whether it exists at all for the human Polε complex remains to be verified; finally, where POLE3-POLE4 structurally engage with full human Polε remains completely unknown.

Here we initially defined by CRISPR genetic screens a minimum set of genes required for survival of POLE3-POLE4 KO cells. From this analysis, iron metabolism factors and the CHTF18-RFC complex emerged as crucially required to support viability of POLE3-POLE4 KO cells. By combining cell biology, biochemistry and structural biology methods, we demonstrate the essential role played by an Iron Sulfur Cluster in the catalytic subunit of Polε in DNA synthesis and genome stability. Furthermore, single molecule DNA fibers, biochemistry and structural modelling support the existence of two tiers of regulation of DNA synthesis by Polε; the first one relies on CHTF18-dependent PCNA loading at leading strands, while the second depends on dsDNA "gripping" at the end of PCNA-encircled ssDNA-dsDNA junction by POLE3-POLE4. Loss of these two tiers of regulation is incompatible with productive leading strand synthesis and cell viability.

## Results

### A map of the genetic dependencies of POLE4 KO cells

To unveil the genetic determinants of cell fitness in a POLE4-deficient background, we interrogated the human genome by CRISPR-Cas9 genetic screens. To this end, we infected RPE1 p53-/- and HeLa Trex Flip In (hereafter HTF) WT and KO for POLE4, engineered to stably express Cas9, with a lentiviral whole-genome sgRNA library (Fig. 1A and Material and Methods). Cells were selected with puromycin and grown for additional 6-7 days or 13-14 days before genomic DNA was extracted for PCR, Illumina sequencing and sgRNA enrichment evaluation (Fig. 1A). After MaGeCK analysis we reconstructed a genetic map of the factors required to support viability of POLE4 KO but not WT cells (Fig. 1B-C, Supplementary Fig. 1 and Supplementary Data 1). Not surprisingly, the most significant hits from our screens were genes involved in genome stability (Fig. 1B, red dots). However, in addition to this, a subset of genes required for the control of iron metabolism scored significantly in RPE1 p53-/- and, to a higher extent, HTF POLE4 KO cells (Fig. 1B, blue dots). We then performed a STRING analysis of the top hits scoring in either cell line. The resulting network revealed closely clustered components of alternative clamp loader complexes shared between RPE1 p53-/- and HTF POLE4 KO cells (Fig. 1C); among them, members of the CHTF18-RFC complex scored as the most significant hits required for viability of POLE4 KO cells in both cellular backgrounds. In addition to this, components of the RAD9-RAD1-HUS1 complex clustered as highly significant in RPE1 p53-/- but not HTF, while multiple members of the Fanconi Anaemia complex were significantly enriched in HTF. This phenomenon might depend on different requirements of these genes for viability (e.g. Fanconi anaemia

genes being essential in RPE1 p53-/-) or the activation of different DNA repair pathways (e.g. activation of the 9-1-1 complex vs the Fanconi anaemia pathway in RPE1 p53-/-) in response to DNA lesions generated by the absence of POLE4 (Fig. 1C). The top hit among iron metabolism genes, scoring in both RPE1 p53-/- and HTF POLE4 KO cells, was NCOA4 (Nuclear Receptor Co-activator 4), an essential autophagic cargo receptor required for ferritin degradation and iron homeostasis[17,18]. In addition to this, PCBP1, a major cytosolic iron chaperone required for iron deposition into ferritin, scored as significant hit in RPE1 p53-/- (Fig. 1C), suggesting that iron storage and release from ferritin plays a crucial role in supporting viability of POLE4 KO cells[19]. Finally, other important regulators of iron metabolism such as STEAP3 and IREB2/IRP2 (Iron Responsive Element Binding Protein 2), appeared to be specifically required for viability of HTF POLE4 KO cells. All together our screens defined a genetic map of the factors required to support the viability of POLE4 KO cells and effective Polε function.

### POLE3 and POLE4 KO cells require a strict control of iron homeostasis for viability

To understand the interplay between POLE3-POLE4 and iron metabolism genes we initially focused on NCOA4, which scored among top hits in both RPE1 p53-/- and HTF POLE4 KO cells, and is required to maintain stable intracellular $Fe^{2+}$ levels to sustain iron-dependent cellular processes via ferritinophagy[17,18]. In order to validate the synthetic lethality between loss of POLE3-POLE4 and NCOA4, we transfected HTF and RPE1 p53-/- with siRNAs against NCOA4 or a CTR and measured proliferation by IncuCyte live-cell imaging. In agreement with our CRISPR screening data, silencing of NCOA4 almost completely abolished proliferation of POLE3 and POLE4 KO cells, in both the HTF and RPE1 p53-/- cellular backgrounds (Fig. 2A, B). To understand the mechanism behind this phenomenon, we analysed cell cycle and EdU incorporation profiles of POLE3-POLE4 KO cells, 48 hours after siRNA transfection, by flow cytometry. Strikingly, silencing of NCOA4 in POLE3-POLE4 KO, but not WT, cells caused a strong increase in the percentage of cells in the G2/M phases of the cell cycle, pointing to the accumulation of replicative DNA damage (Fig. 2C, D). Consistently with this, silencing of NCOA4 was associated with a considerable increase in γH2AX levels, specifically in the G2/M phases, as measured by FACS (Fig. 2D, E). Importantly, knock-down of NCOA4 also caused a significant reduction of fork extension rates in POLE3 and POLE4 KO cells, pointing to a defect in leading strand synthesis (Supplementary Fig. 2A). In addition to NCOA4, the iron responsive protein IREB2/IRP2 scored as a top hit in HTF but not RPE1 p53-/- POLE4 KO cells (Fig. 1B, C). Consistently with this, transient silencing of IREB2/IRP2 by siRNA led to reduced proliferation of POLE3 and POLE4 KO cells as measured by IncuCyte Live-cell imaging (Fig. 2F). In agreement with our screen, this effect was less evident and not statistically significant in RPE1 p53-/- POLE3 and POLE4 KO cells (Supplementary Fig. 2B), which is likely caused by redundancy and different expression levels of IRP1 and IRP2. Finally, to further confirm an iron-dependent mechanism at the basis of these genetic interactions, we treated HTF and RPE1 p53-/- WT and POLE3-POLE4 KO with increasing concentration of the iron chelator deferoxamine (DFO) and analysed viability and cell growth. Consistently with an iron-dependent phenomenon, iron chelation was associated with reduced proliferation of POLE3 and POLE4 KO cells as measured by both colony forming assay and IncuCyte live-cell imaging (Fig. 2G, Supplementary Fig. 2C and Supplementary Fig. 2D). Furthermore, treatment with 3 μM DFO caused a significant reduction in fork extension rates in POLE4 KO cells, suggesting defective leading strand synthesis (Supplementary Fig. 2E). Of note, treatment with DFO caused a small but significant decrease in the levels of POLE1, which was particularly evident in POLE3-POLE4 KO cells; this data suggest that reduced stability of Polε might play at least a partial role in the observed synthetic lethality with iron metabolism factors (Fig. 2H).

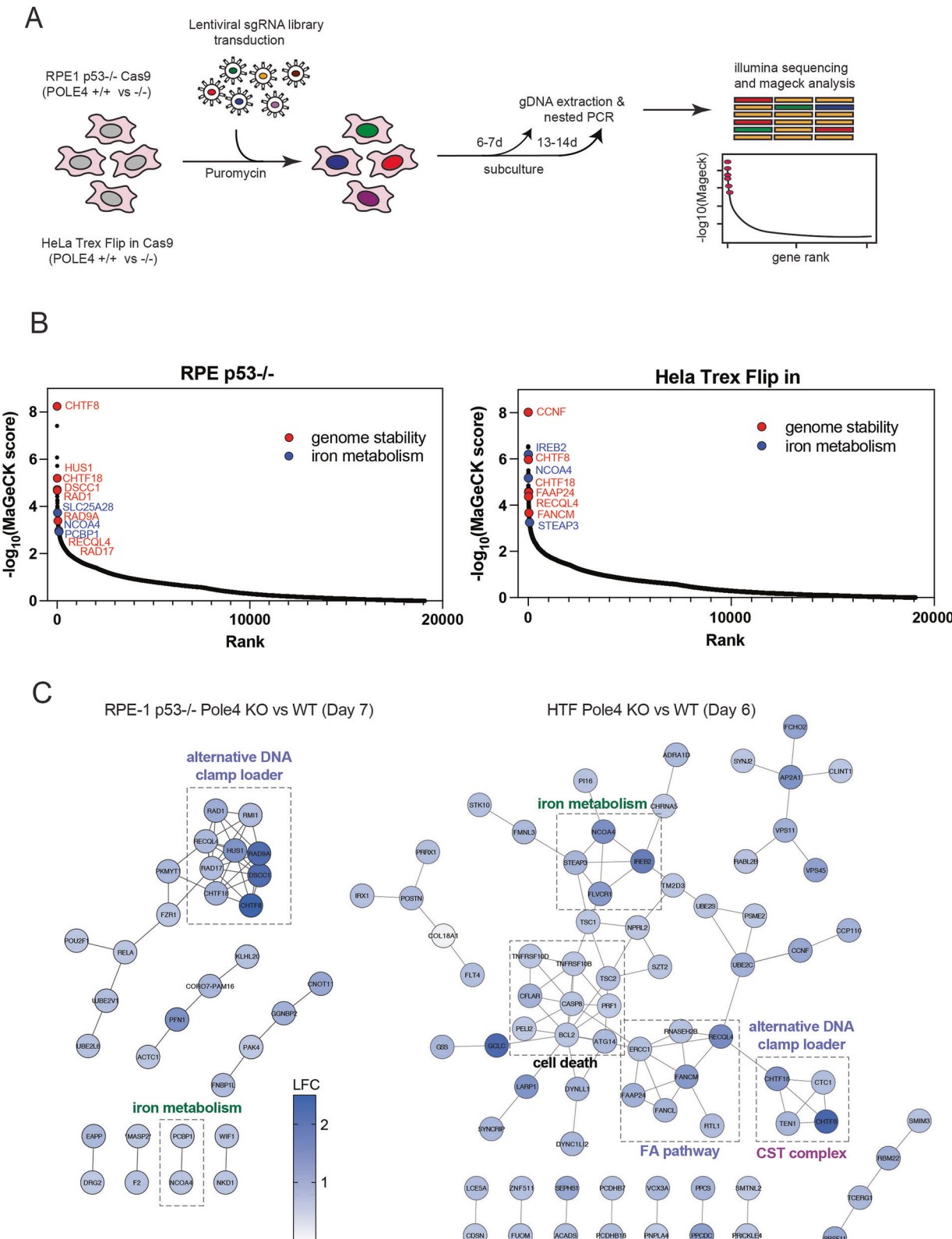

**Fig. 1 | CRISPR screenings reveals genetic dependencies of POLE4 KO cells.**
**A** Schematic of the CRISPR screening pipeline. **B** Gene ranking from MAGeCK score analysis of HeLa Trex Flip In (HTF) and RPE1 p53-/- POLE4 KO vs WT cells, 6 and 7 days after puromycin selection. **C** STRING analysis of interactions between statistically significant hits in HTF and RPE1 p53-/- POLE4 KO cells.

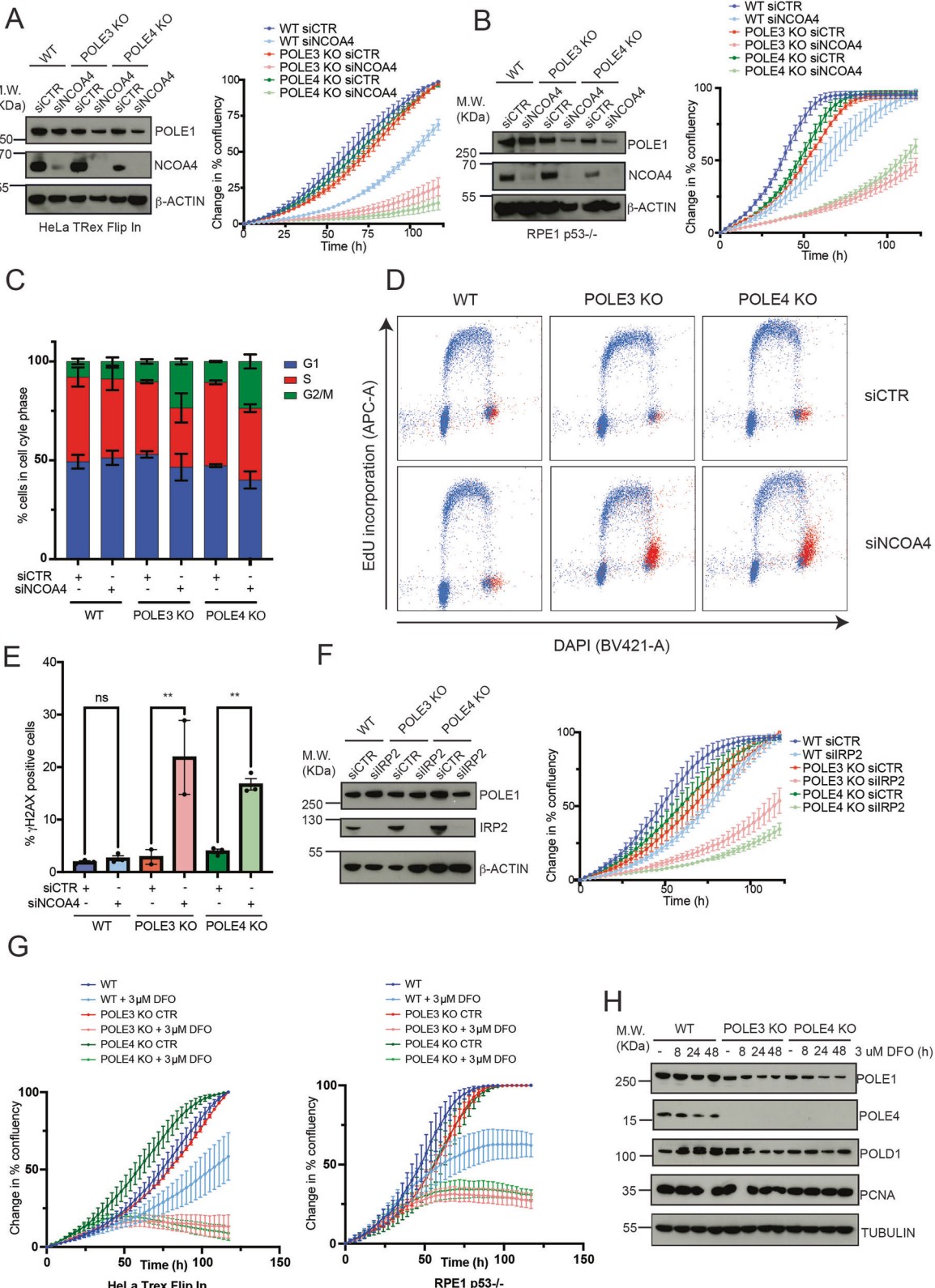

## Mutation of two cysteine residues in the Iron Sulfur Cluster of human Polε abrogates its Polymerase but not DNA binding activity

Several factors involved in the control of genome stability harbour iron sulfur clusters (ISCs). However, recent work from the Possemato lab has suggested that defective biogenesis of ISCs, as induced by loss of NFS1, causes genome instability by specifically impairing Polε

function[20]. This finding and our data suggest that Polε activity requires a tight control of iron homeostasis and that genome instability caused by iron depletion might primarily depend on defective leading strand synthesis. Importantly, previous work in *S. Cerevisiae* has revealed the existence of a functional ISC in the Polymerase domain of budding yeast Polε[21]; this has been recently observed also in cryoEM structures that include the human Polε catalytic domain[14,15]. To evaluate the

**Fig. 2 | POLE3-POLE4 KO cells require a tight control of iron homeostasis for survival. A** Left: western blot analysis of NCOA4 and POLE1 levels in HTF WT, POLE3 KO and POLE4 KO cells transfected with siRNAs against NCOA4 or CTR. β-ACTIN was used for normalization. Right: Change in % confluency of HTF WT, POLE3 KO and POLE4 KO cells transfected with siRNAs against NCOA4 or CTR and imaged every 3 hours for 120 hours by IncuCyte Live-cell imaging; results are reported as mean +/- SEM of triplicate biological experiments. **B** Same experiments than A) in RPE1. **C** Bar graphs showing the percentage of cells in the indicated cell cycle phases as measured by FACS cell cycle analysis. Results are reported as mean +/- SEM of triplicate biological experiments. **D** Representative FACS plots of HTF WT, POLE3 and POLE4 KO cells transfected with siRNAs against NCOA4 or CTR and stained for EdU, DAPI and γH2AX. Red dots correspond to γH2AX positive cells. **E** Bar graphs indicating the percentage of cells positive for γH2AX from FACS cell cycle analysis; results are reported as mean +/- SEM of triplicate biological experiments; one-way ANOVA with Tukey's post-hoc test, n.s. not significant; **$p < 0.01$. **F** Left: western blot analysis of IRP2 and POLE1 levels in HTF WT, POLE3 KO and POLE4 KO cells transfected with siRNAs against IRP2 or CTR. β-ACTIN was used for normalization. Right: Change in % confluency of HTF WT, POLE3 KO and POLE4 KO cells transfected with siRNAs against IRP2 or CTR and imaged every 3 hours for 120 hours by IncuCyte Live-cell imaging; results are reported as mean +/- SEM of triplicate biological experiments. **G** Change in % confluency of HTF (left) and RPE1 p53-/- (right) WT, POLE3 KO and POLE4 KO cells treated with 3 µM Deferoxamine (DFO) and imaged every 3 hours for 120 hours by IncuCyte Live-cell imaging; results are reported as mean +/- SEM of triplicate biological replicates. **H** Western blot analysis of POLE1, POLE4, POLD1 and PCNA levels in HTF WT, POLE3 KO and POLE4 KO treated or not with 3 µM DFO for the indicated times. TUBULIN was used for normalization. Experiments were repeated at least 3 times; a representative experiment is reported here.

functional relevance of this domain in the human counterpart, we expressed and purified from baculovirus-infected insect cells WT human Polε and a mutant harbouring cysteine to serine substitutions (POLE1 C651S/C654S) in two critical residues of the putative ISC domain (Fig. 3A, B). We initially analysed DNA synthesis of WT and ISC mutant Polε by primer extension assays performed on preassembled substrates constituted by a fluorescent 50mer annealed to a ssDNA 80-mer (Fig. 3C, top). Strikingly, while WT Polε rapidly and fully extended this substrate in vitro, the ISC mutant was completely incompetent for DNA synthesis (Fig. 3C, bottom), pointing to a crucial role for Polε ISC in sustaining its enzymatic polymerase activity. Importantly, exonuclease assays performed on the same substrates, in the absence of dNTPs, revealed that the ISC mutant is also lacking a functional exonuclease activity (Fig. 3D). DNA Polymerase Delta (Polδ), the lagging strand Polymerase, also harbours an ISC in its catalytic domain. Lack of this domain has been recently reported to compromise its DNA synthesis and exonuclease activities in association with loss of DNA binding[22]. To verify if this was the case for human Polε, we performed a set of EMSAs (Electrophoretic Mobility Shift Assays) with WT and ISC mutant Polε using different fluorescent substrates, comprising ssDNA, dsDNA and a replication fork structure mimicking a leading strand (Fig. 3E top panels). Strikingly, at a difference from human Polε, loss of the ISC of Polε did not significantly affect binding to ssDNA, dsDNA or, more importantly, leading strands mimicking structures (Fig. 3E). Overall, our in vitro and in vivo data suggest that human cells require a strict control of iron homeostasis to support ISC-dependent leading strand synthesis by Polε. Loss of iron coordination by Polε does not significantly affect DNA binding but severely compromises its polymerase and exonuclease activities.

## A profound synthetic lethality between the loss of POLE3-POLE4 and the CHTF18-RFC complex

In vitro reconstitution of DNA replication with purified human proteins has recently revealed a crucial role for the CHTF18-RFC complex in sustaining efficient human leading strand DNA synthesis[4]. More recently, the structural basis for CHTF18-dependent PCNA loading, specifically at leading strands, has been revealed by using either budding yeast[23] or human proteins[24]. However, if and how loss of CHTF18 impacts replication fork rates and genome stability in human cells remains to be verified. With this in mind, we generated CHTF18 KO cells in the HTF cellular background by CRISPR Cas9-based editing. Strikingly, while we obtained several clones lacking CHTF18 in a HTF "WT" background, no clone in a POLE3 and POLE4 KO setting showed loss of CHTF18, pointing to a *bone fide* synthetic lethal interaction between these replisome factors. Of note, while CHTF18 KO cells appeared to grow similarly to the WT background in culture, when seeded in low confluency for IncuCyte Live-cell imaging, we could note a slightly slower proliferation rate (Fig. 4A). In order to explore the mechanistic basis for the synthetic lethality between loss of CHTF18-RFC and POLE3-POLE4 we analysed the consequence of transient

silencing of POLE3-POLE4 in CHTF18 KO cells and *viceversa*. In agreement with our CRISPR screen data and experimental observations, transient silencing of POLE3-POLE4 caused a remarkable impairment in the proliferation of CHTF18 KO cells (Fig. 4B and C). This data suggests that, while proficient in cellular proliferation in the unchallenged condition, CHTF18 KO cells are strictly dependent on POLE3-POLE4 function for viability. Importantly, a similar effect was obtained by transient knockdown of CHTF18 in POLE3 and POLE4 KO cells (Fig. 4D, E). Finally, to confirm the involvement of CHTF18, specifically in the context of the CHTF18-RFC complex, we silenced by siRNAs the other components of the complex, CHTF8 and DSCC1, for which no functional antibodies are currently available. In agreement with our screen data, silencing of CHTF8 and DSCC1, as validated by quantitative real time PCR (qPCR), led to reduced viability of POLE3-POLE4 KO cells but not their WT counterparts (Supplementary Fig. 3A-B-C-D). In order to understand the molecular basis of this synthetic lethal interaction, we then analysed the dynamic of DNA replication by DNA fiber assay, in POLE3-POLE4 KO cells, in basal conditions and upon silencing of CHTF18 and *viceversa*. Strikingly, silencing of CHTF18, in POLE3-POLE4 KO cells, caused a remarkable reduction of fork extension rates pointing to defective DNA synthesis (Fig. 4F). Of note, this was associated to increased fork stalling events, suggesting a replication stress condition (Fig. 4G). Interestingly, and in agreement with a role for CHTF18 in regulating PCNA loading at the leading strand and Polε processivity, CHTF18 KO cells presented with a significant reduction in fork speed in basal conditions and a modest but significant increase in fork asymmetry (Fig. 4H, I). However, in agreement with the observed synthetic lethality, silencing of POLE3-POLE4 caused a further reduction of fork extension rates in CHTF18 KO cells, in association with increased fork stalling (Fig. 4H, I). In summary our data reveal that POLE3-POLE4 and the CHTF18-RFC complex represent two essential tiers of regulation of leading strand synthesis in vivo. Loss of both these activities is incompatible with productive DNA replication and cell viability.

## The structural basis of POLE3-POLE4 engagement with POLE1

While a structure of full human Polε is still missing, structural work on the budding yeast proteins has suggested that the orthologs of POLE3-POLE4 interact with a 'mooring helix' in the linker between the catalytic and non-catalytic domains and also make additional interactions with both domains, acting as a wedge between the catalytic and non catalytic modules of Polε to confer a more stable conformation to the complex[16]. In the absence of experimental structures for the human complex, to understand how POLE3-POLE4 might regulate Polε stability and DNA synthesis we turned to AlphaFold3 to predict what interactions the accessory subunits make with the rest of the Polε complex (Fig. 5A and Supplementary Fig. 4A). While the interactions with the 'mooring helix' appeared to be conserved, a structure corresponding to the rigid yeast structure was not observed, with instead POLE3-POLE4 anchored to the linker but fully flexible to access all

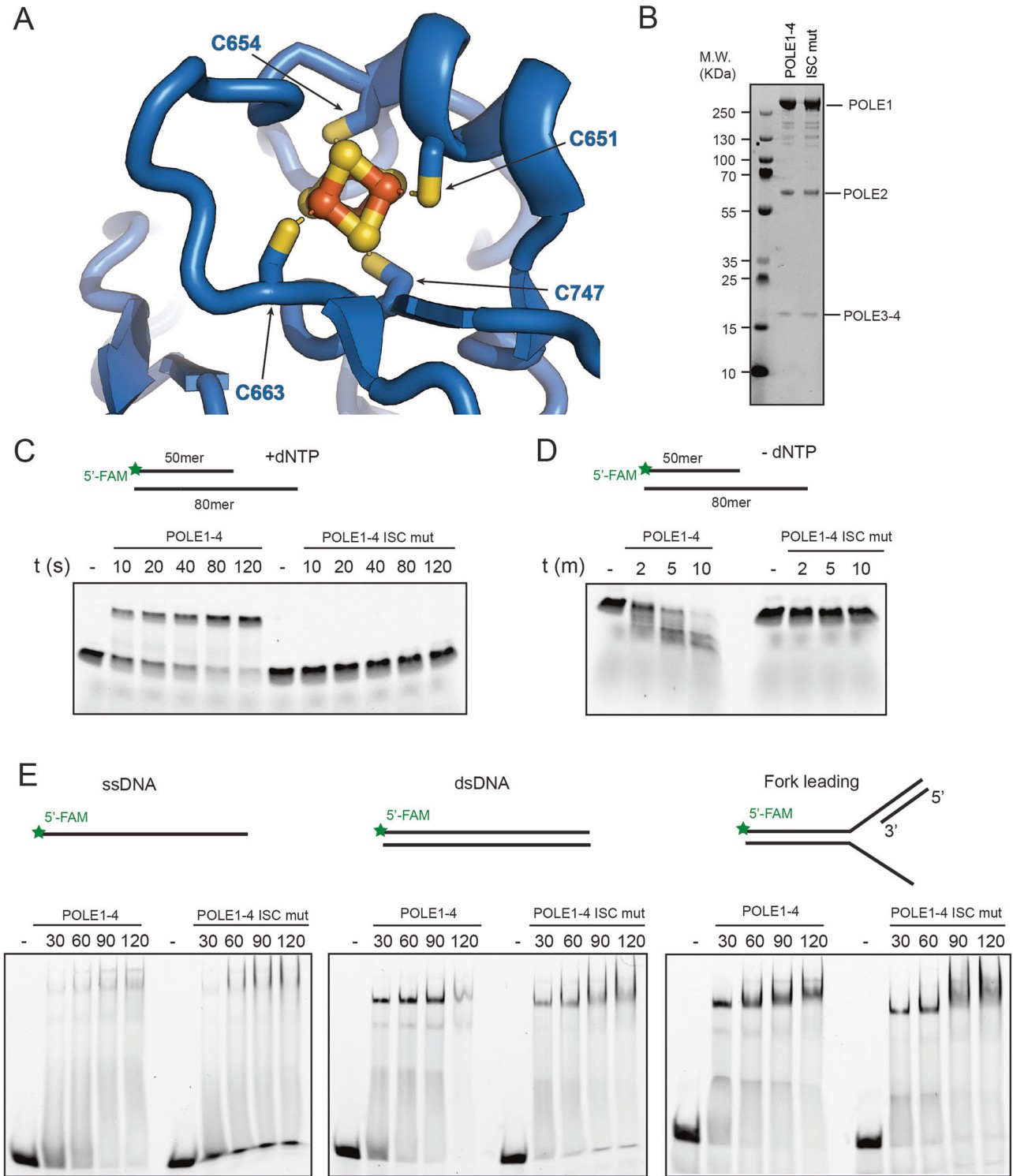

**Fig. 3 | A Polε ISC mutant is defective in DNA synthesis and exonuclease activities. A** Cartoon representation of the Fe-S cluster of Polε, with key coordinating cysteine side chains represented as sticks, produced using PDB:9B8T. **B** Coomassie staining of purified Polε WT and Iron Sulphur Cluster (ISC) mutant (C651S/C654S). **C** Top: representative scheme of the fluorescently labelled substrate used in the primer extension assays; Bottom: representative denaturing UREA gel from primer extension assays: Polε WT and ISC mutant were incubated for the indicated times in the presence of equimolar amount of fluorescent substrates and 0.1 mM dNTP. Experiments were repeated 3 times; a representative experiment is shown here. **D** Top: representative scheme of fluorescently labelled substrate used

in the exonuclease assays; dNTPs were excluded from the reactions. Bottom: denaturing UREA gel from exonuclease assays: Polε WT and ISC mutant were incubated for the indicated times in the presence of equimolar amount of fluorescent substrates. Experiments were repeated 3 times; a representative experiment is shown here. **E** Top: representative scheme of fluorescently labelled substrate used in the exonuclease assays. Bottom: Binding of Polε WT and ISC mutant to ssDNA (left); dsDNA (middle) and leading strand (right) analysed by EMSA; increasing amount of proteins were incubated with the indicated substrates for 15 min on ice; reactions were finally run on native TBE gels and imaged using a phosphorimager. Experiments were repeated 3 times; a representative experiment is shown here.

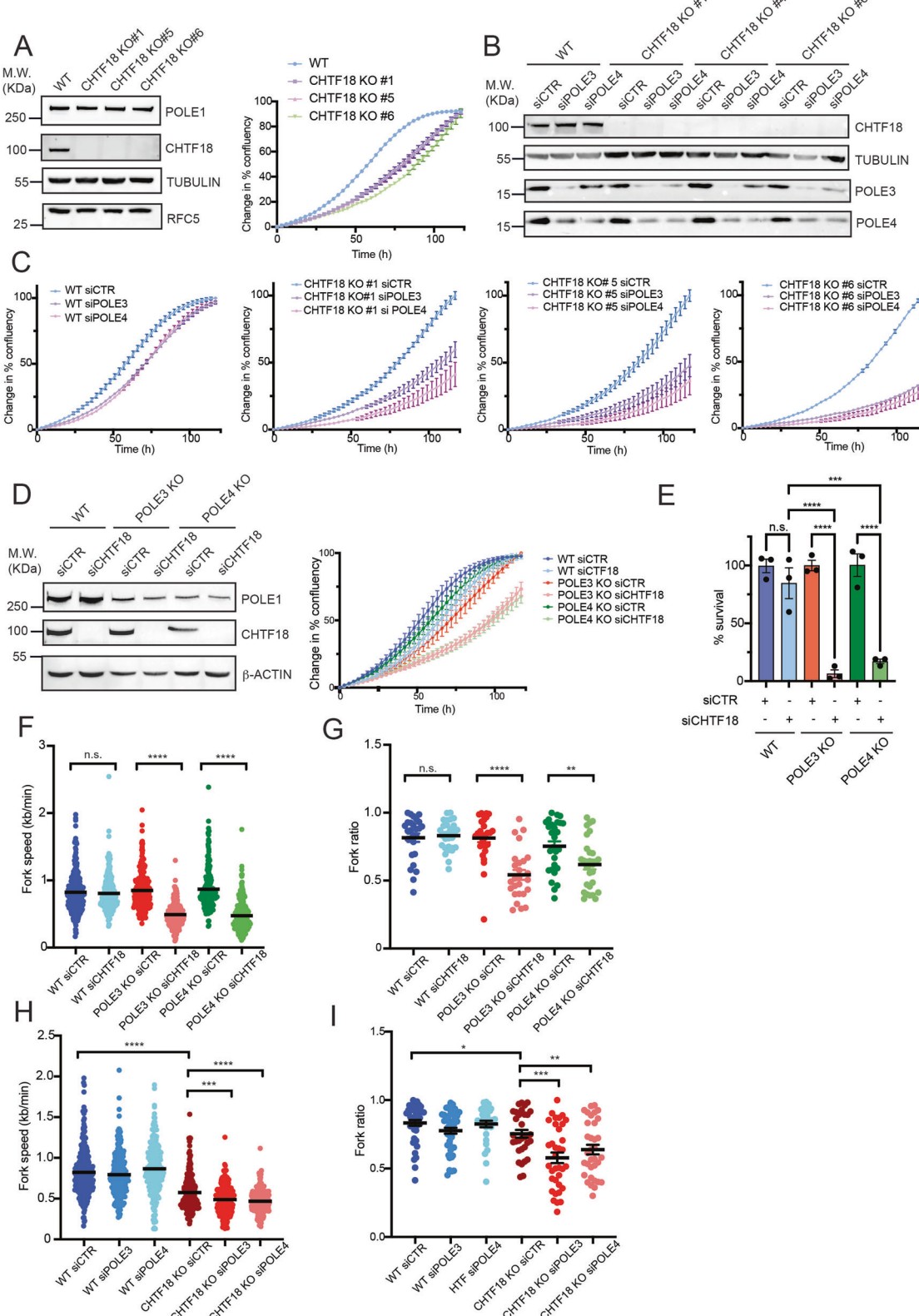

conformations within the reach of the linker, with most models favouring positioning adjacent to the non-catalytic domain of POLE1 without having a clearly defined interface. Consistently with our AlphaFold3 model cross-linking mass spectrometry of full Polε revealed cross-links between the linker region near the putative mooring helix and both POLE3 and POLE4 (Fig. 5B and Supplementary Data 2). Additional crosslinks between POLE3-POLE4 and the N and C

terminal domains of POLE1 could not be satisfied by crosslinks between the corresponding residues in the rigid yeast structure (Supplementary Fig. 4B) but could be accounted for by the flexibility in the position of POLE3-POLE4 in the AlphaFold3 predictions. To further confirm the structural model in human cells, we generated a series of Flag-tagged versions of POLE1, POLE3 and POLE4 and tested interaction in vivo with endogenous proteins by co-immunoprecipitation in

**Fig. 4 | POLE3-POLE4 KO cells require CHTF18-RFC for survival. A** Left: western blot analysis of CHTF18, POLE1 and RFC5 levels in HTF WT and CHTF18 KO cells. TUBULIN was used for normalization. Right: Change in % confluency of HTF WT and CHTF18 KO cells (clone #1, #5 and #6) imaged every 3 hours for 120 hours by IncuCyte Live-cell imaging; results are reported as mean +/- SEM of triplicate experiments. **B** Western blot analysis of CHTF18, POLE3 and POLE4 levels in HTF WT and CHTF18 KO (clone #1, #5 and #6) transfected with siRNAs against POLE3-POLE4 or CTR. TUBULIN was used for normalization. Experiments were repeated 3 times; a representative WB is shown here. **C** Change in % confluency of HTF WT and CHTF18 KO cells (clone #1, #5 and #6) transfected with siRNAs against POLE3, POLE4 or CTR and imaged every 3 hours for 120 hours by IncuCyte Live-cell imaging; results are reported as mean +/- SEM of triplicate biological experiments. **D** Left: western blot analysis of CHTF18 and POLE1 levels in HTF WT, POLE3 KO and POLE4 KO cells transfected with siRNAs against CHTF18 or CTR. TUBULIN was used for normalization. Right: Change in % confluency of HTF WT, POLE3 KO and POLE4 KO cells transfected with siRNAs against CHTF18 or CTR and imaged every 3 hours for 120 hours by IncuCyte Live-cell imaging; results are reported as mean +/- SEM of triplicate biological experiments. **E** Bar graphs showing the percentage of clonogenic survival of HTF WT and POLE3-POLE4 KO cells transfected with siRNA against CHTF18. Results are reported as mean +/- SEM, are normalized to siRNA CTR-

transfected cells and were obtained from 3 independent biological experiments. One way ANOVA with Sidak's post-hoc test; ****$p < 0.0001$, n.s. not significant. **F** Bar graphs showing replication fork extension rates from HTF WT, POLE3 and POLE4 KO transfected with siRNA against CHTF18 or control; Results are reported as mean +/- SEM; unpaired one sided t-test analysis ****$p < 0.0001$, n.s. not significant. At least 200 fibers were used for each condition for the analysis (**G**). Bar graphs showing the ratio of shorter vs longer replication forks at either side of newly activated replication origins, from HTF WT, POLE3 and POLE4 KO cells transfected with siRNA against CHTF18 or control; Results are reported as mean +/- SEM; unpaired one sided t-test analysis ****$p < 0.0001$, **$p < 0.01$; n.s. not significant. At least 16 fibers were used for each condition for the analysis (**H**). Bar graphs showing replication fork extension rates from HTF WT, and CHTF18 KO cells transfected with siRNA against POLE3, POLE4 or control; Results are reported as mean +/- SEM; unpaired one sided t-test analysis ****$p < 0.0001$, n.s. not significant. At least 200 fibers were used for each condition for the analysis. **I** Bar graphs showing the ratio of shorter vs longer replication forks at either side of newly activated replication origins, from HTF WT and CHTF18 KO cells transfected with siRNA against POLE3, POLE4 or control; Results are reported as mean +/- SEM; unpaired one sided t-test analysis ****$p < 0.0001$, **$p < 0.01$; n.s. not significant.

---

293 T cell extracts. Consistently with our model, mutation of the two critical tryptophan residues in the "mooring" helix of human Polε (W1271 and W1279) abrogated interaction of Flag-POLE1 with endogenous POLE3, while mutation of V43 in POLE3 and D53 in POLE4 led to a similar loss of interaction between Flag-POLE3 and Flag-POLE4 with endogenous POLE1 (Fig. 5C, D, E). A less prominent, effect was observed by breaking a salt bridge on the interface by mutating E77 in POLE3 (Fig. 5C, D, E). Importantly, re-expression of WT POLE4 but not the D53K mutant in POLE4 KO cells rescued the synthetic lethality and fork defect associated with silencing of CHTF18, pointing to a replication-specific defect (Supplementary Fig. 5A and Supplementary Fig. 5B).

Overall our data suggest that POLE3-POLE4 interacts with POLE1 via a conserved human "mooring" helix and this conformation might help promote Polε stability and function.

## Loss of POLE3-POLE4 impairs dsDNA binding and DNA synthesis of full Polε

Work in vitro with purified budding yeast proteins has previously showed that Dpb3 and Dpb4 are required to promote full Polε processivity[25]. Furthermore, we and others have previously shown that Dpb3-Dpb4 and POLE3-POLE4 are bona fide DNA binding proteins[5,26]. However, if and how POLE3-POLE4 affect DNA synthesis and DNA binding of full human Polε remains unknown. With this in mind, we expressed and purified a Polε complex lacking POLE3-POLE4 and analysed its biochemical activities using the previously described assays (Fig. 6A). Initially, we tested in vitro the ability of a POLE1-POLE2 subcomplex to extend DNA substrates by primer extension assays. Consistently with a role for POLE3-POLE4 in supporting DNA synthesis of Polε, the efficiency in extending such substrates was significantly lower, when POLE1-POLE2 was compared to full Polε (Fig. 6B). Importantly, the POLE1-POLE2 subcomplex presented reduced efficiency in its exonuclease activity when compared to full Polε, all together pointing to reduced engagement with its substrate (Fig. 6C). We then analysed binding to DNA in vitro by EMSA, using the previously described substrates. Strikingly, while loss of POLE3-POLE4 did not affect binding to ssDNA, a Polε complex lacking POLE3-POLE4 was almost completely unable to bind dsDNA (Fig. 6D). Importantly loss of POLE3-POLE4 was associated with reduced affinity for a replication fork structure mimicking a leading strand, pointing to reduced engagement with its physiologic substrate (Fig. 6D). Of note, in the leading fork EMSAs we could observe the appearance of a double band in the shift assays in the POLE1-POLE2 only lanes, suggesting that two molecules of Polε might abnormally engage with this substrate in these experimental settings. All together our data suggest that POLE3-POLE4

support DNA synthesis of full Polε by favouring engagement with dsDNA at replication forks.

## Loss of dsDNA binding by POLE4 lead to defective DNA synthesis

How POLE3-POLE4 might engage with dsDNA to support Polε-dependent DNA synthesis in the absence of CHTF18-RFC-loaded PCNA remain unknown. To answer this question we used AlphaFold3 to model Polε bound to a DNA molecule equivalent to an elongating substrate (Fig. 7A and Supplementary Fig. 6A). The attachment of POLE3-POLE4 to the 'mooring helix' in the linker region allows sufficient flexibility for the accessory subunits to reach the newly synthesised DNA exiting the polymerase. This crucially enables POLE3-POLE4 to make further interactions with the nascent dsDNA, interacting non-specifically, using positively charged side chains to balance the negative charge of the DNA backbone, and bending the DNA through interactions with their histone like folds (Fig. 7B). To validate this model we generated and expressed in insect cells a mutant form of the full Polε complex containing POLE4 with predicted impaired DNA binding through steric repulsion and charge reversal (A44L, R45D, K47E, K51E) (Fig. 7C). Consistently with our model, mutation of these POLE4 residues abolished binding to dsDNA but not ssDNA and was associated with a significant reduction in binding to leading strands at replication forks (Fig. 7D). Importantly, and consistently with our model, loss of dsDNA binding was associated with reduced DNA synthesis (Fig. 7E) and exonuclease activities (Fig. 7F). Furthermore, a Polε complex containing POLE4 DBM was still able to interact with histone H3-H4, pointing to a separation of function mutant (Supplementary Fig. 7). Importantly expression of this mutant in cells did not rescue the synthetic lethality with silencing of CHTF18 as WT POLE4 (Supplementary Fig. 5A and Supplementary Fig. 5B). Finally, to understand how POLE3-POLE4 and CHTF18-dependent PCNA loading co-operates to promote full Polε activity, we used AlphaFold3 to model Polε bound to both DNA and PCNA (Fig. 7G and Supplementary Fig. 6B). In these models, despite regions of the linker interacting with PCNA, POLE3-POLE4 are still able to make similar interactions with dsDNA protruding from behind the clamp, suggesting that in the presence of PCNA the accessory subunits might further enhance the dsDNA binding and processivity of the polymerase. To test the interplay between CHTF18-RFC and POLE3-POLE4 in full processivity assays, we finally performed radioactive assays on ssDNA primed M13 substrates, reaching 7-8 Kb in length. In accordance with previous work, when incubated with these substrates both full Polε and POLE1-POLE2 alone were minimally able to effectively synthesize DNA[4]. Importantly, addition of CHTF18-RFC strongly increased DNA synthesis rates of Polε pointing to a crucial role for CHTF18-RFC in Polε-

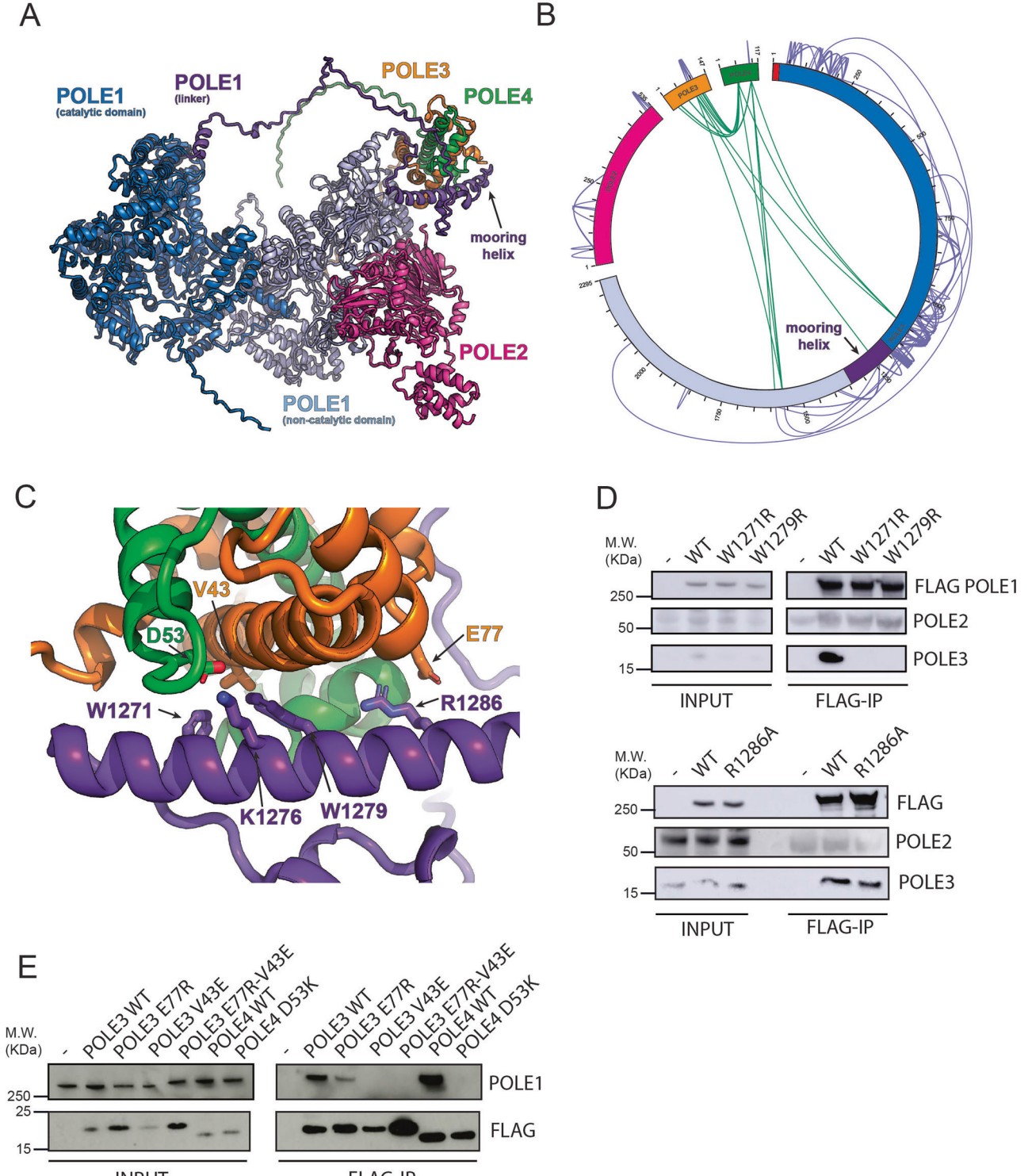

**Fig. 5 | POLE3-POLE4 bind POLE1 via a conserved mooring helix. A** Cartoon representation of the AlphaFold3 model of Polε. The catalytic, linker and non-catalytic domains of POLE1 are shown in shades of blue and purple while POLE2, POLE3 and POLE4 are coloured magenta, orange and green respectively. **B** Map of DSBU cross-links identified within full Polε complex. POLE1 vs POLE3-POLE4 cross-links are shown in green. **C** Close up view of the predicted POLE1 mooring helix interactions with POLE3 and POLE4 subunits, key sidechains are displayed as sticks. **D** Western blot analysis of FLAG-immunoprecipitations from 293 T cells transiently transfected with vectors expressing WT FLAG-POLE1 or described POLE1 mutants. Immunoprecipitation experiments were repeated at least two times; a representative experiment is shown here. **E** Western blot analysis of FLAG-immunoprecipitations from 293 T cells transiently transfected with vectors expressing WT FLAG-POLE3, WT FLAG-POLE4 or described POLE3-POLE4 mutants. Immunoprecipitation experiments were repeated at least two times; a representative experiment is shown here.

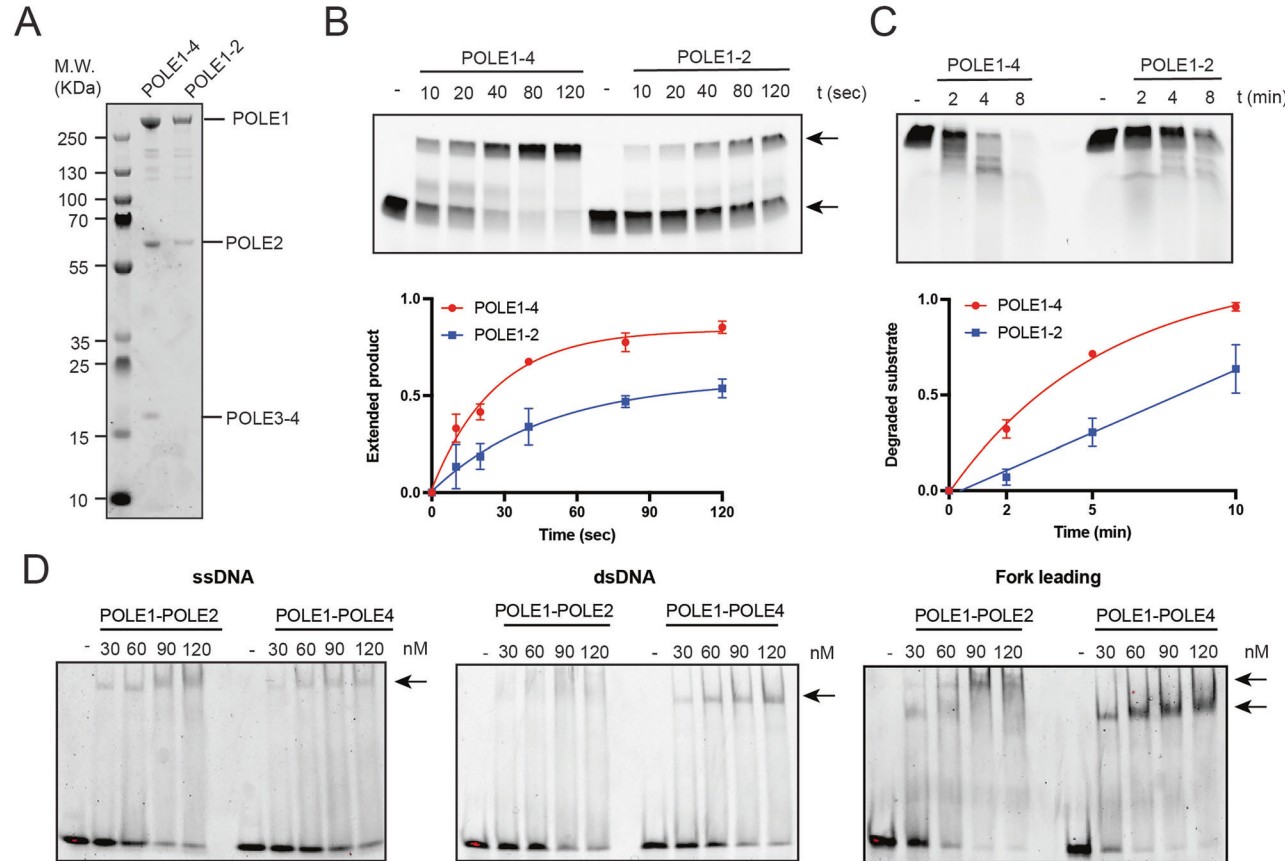

**Fig. 6 | POLE3-POLE4 aid polymerase and nuclease activity of Polε. A** Coomassie staining of purified full Polε (POLE1-POLE2-POLE3-POLE4) or POLE1-POLE2 only (**B**) Top: Representative denaturing UREA gel from primer extension assays; full Polε or POLE1-POLE2 only were incubated for the indicated times in the presence of equimolar amount of fluorescent substrates and 0.1 mM dNTPs; Bottom: quantification of triplicate primer extension assays performed with full Polε or POLE1-POLE2 only. Results are reported as mean +/- SD. **C** Top: Representative denaturing UREA gel from exonuclease assays; full Polε or POLE1-POLE2 only were incubated for the indicated times in the presence of equimolar amount of fluorescent substrates in the absence of dNTP; Bottom: quantification of triplicate exonuclease assays performed with full Polε or POLE1-POLE2 only. Results are reported as mean +/- SD. **D** Binding of full Polε or POLE1-POLE2 only to ssDNA (left); dsDNA (middle) and leading strand (right) analysed by EMSA; increasing amount of proteins were incubated with the indicated substrates for 15 min on ice; reactions were finally run on native TBE gels and imaged using a phosphorimager. Experiments were repeated at least 3 times; a representative experiment is shown here.

dependent DNA synthesis, in accordance with recently published work[4] (Fig. 7H). However and strikingly, when CHTF18-RFC was incubated with POLE1-POLE2 alone, the extension rate of ssDNA primed substrates was strongly reduced compared to the full polymerase complex (Fig. 7H). Importantly, a similar result was obtained when comparing the extension rates of WT Polε vs Polε DNA binding mutant, in the presence of CHTF18-RFC, pointing to a DNA binding-specific effect (Supplementary Fig. 8A). Consistently with a PCNA loading-dependent mechanism, addition of CHTF18-RFC alone did not stimulate DNA synthesis by Polε (Supplementary Fig. 8B). All together, this points to a cooperative mechanism of DNA synthesis at replication forks that involves CHTF18-dependent PCNA loading and dsDNA binding by POLE3-POLE4.

## Discussion

Here we initially performed an unbiased whole-genome CRISPR screen to identify synthetic lethal interactions with loss of the non-essential subunits of Polε, POLE3-POLE4. This unbiased approach allowed us to identify a novel function for POLE3-POLE4 in sustaining Polε−dependent DNA synthesis and a set of genes required for efficient leading strand synthesis and genome stability.

In addition to genome stability factors, iron metabolism genes scored among the top hits in our screens in POLE4 KO cells. While pointing to a crucial vulnerability of POLE3-POLE4 KO cells, this data suggests that Polε function is particularly dependent on sustained iron

homeostasis. Consistently with this, transient silencing of the essential ferritinophagy regulator NCOA4, the most conserved among these hits, caused a rapid proliferative block in POLE3-POLE4 KO cells; this was associated with the accumulation of γH2AX positive cells in the G2/M phase of the cell cycle, pointing to the accumulation of replicative DNA damage. Dependency on other iron-regulating factors significantly differed between RPE1 and HTF in our screens and experimental conditions. It is likely that a different dependency on iron regulatory proteins (e.g. IRP1 vs IRP2) and intracellular iron-sustaining processes (e.g. iron uptake vs ferritinophagy) might be responsible for this phenomenon. Despite this, and consistently with an iron-dependent mechanism, treatment of both HTF and RPE1 POLE3-POLE4 KO cells with the iron chelating agent DFO resulted in a rapid block of cellular proliferation as assessed by both CFA and IncuCyte live cell imaging.

Transient silencing of NCOA4 and treatment with DFO were associated with a small but significant reduction of Polε levels in human cells. Since POLE3 and POLE4 KO cells harbour constitutively lower levels of Polε, it is possible that the combinatorial loss of POLE3-POLE4 and ISC biogenesis might bring Polε levels below the threshold required for cellular survival. It is indeed well known that loss of ISC biogenesis leads to a reduction of the levels of ISC-containing proteins[27]. In addition to this, we have also shown that a recombinant Polε lacking 2 essential cysteine residues, required to coordinate an ISC, is incompetent for DNA synthesis in vitro, despite being fully

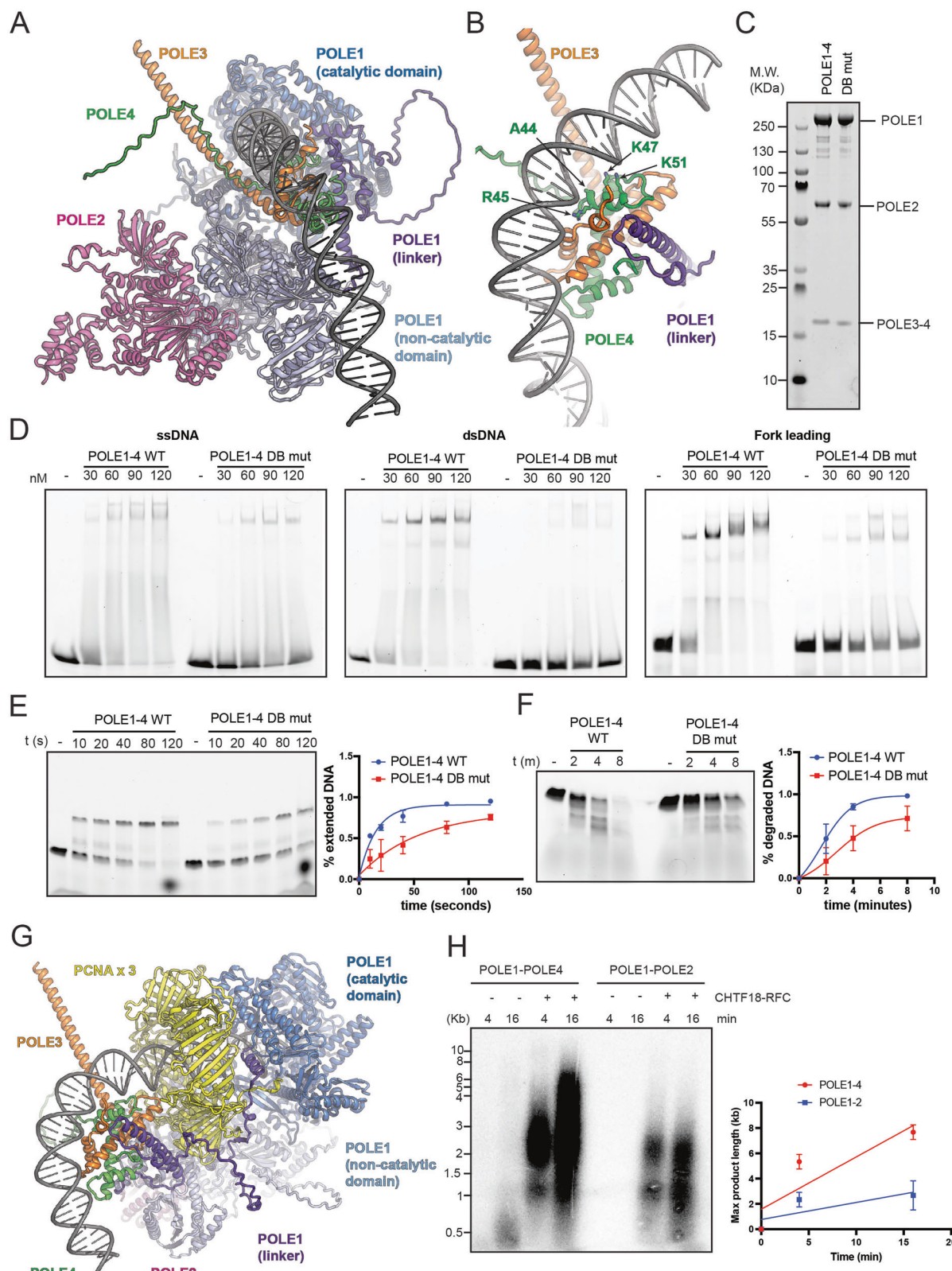

able to engage with its DNA substrate. The ability of the Polε ISC mutant to engage with a DNA substrate but be unable to perform the extension reaction can be explained structurally as it has been proposed that the Fe-S location at the end of the finger in the catalytic domain allows it to act as a pivot as the finger moves while transitioning from the nucleotide bound to nucleotide exchange state as part of the catalytic cycle[14,15]. The presence of Polε lacking the ISC

would therefore be potentially highly deleterious to the cell as it would be able to be incorporated into the CMG, engage with the primer template DNA substrate, but then be unable to synthesise DNA, while blocking access to other polymerases. Thus, a combination of reduced stability and ineffective DNA synthesis might explain the observed genetic interaction between loss of POLE3-POLE4 and iron metabolism factors. Previous work from the Possemato's lab has showed that

**Fig. 7 | POLE3-POLE4 interact with nascent dsDNA. A** Cartoon representation of the AlphaFold3 model of Polε complex with DNA. The catalytic, linker and non-catalytic domains of POLE1 are shown in shades of blue and purple while POLE2, POLE3 and POLE4 are coloured magenta, orange and green respectively, and the DNA is shown in shades of grey. **B** Close up view of the predicted interactions of POLE3 and POLE4 subunits with the dsDNA with key sidechains displayed as sticks. **C** Coomassie staining of purified WT Polε or a Polε mutant harbouring POLE4 mutations that disrupt dsDNA binding (DB mutant). **D** Binding of WT Polε or DB mutant to ssDNA (left), dsDNA (middle) and leading strand (right) analysed by EMSA; increasing amount of proteins were incubated with the indicated substrates for 15 min on ice; reactions were finally run on native TBE gels and imaged using a phosphorimager. Experiments were repeated at least 3 times; a representative experiment is shown here. **E** Left: representative denaturing UREA gel from primer extension assays: WT Polε or DB mutant were incubated for the indicated times in the presence of equimolar amount of fluorescent substrates and 0.1 mM dNTPs. Right: quantification of triplicate primer extension assays performed with WT Polε or DB mutant. Results are reported as mean +/- SD. **F** Left: denaturing UREA gel from exonuclease assays: WT Polε or DB mutant were incubated for the indicated times in the presence of equimolar amount of fluorescent substrates in the absence of dNTP; Right: quantification of triplicate exonuclease assays performed with WT Polε or DB mutant. Results are reported as mean +/- SD. **G** Cartoon representation of the AlphaFold3 model of Polε complex with PCNA and DNA, coloured as for A with additional PCNA shown in yellow. **H** Left: Representative radioactive primer extension assay performed on M13mp18 substrates in the presence of the indicated proteins and for the indicated time points. PCNA and RPA were included in all the reactions. Right: quantification of triplicate radioactive primer extension assays performed with WT Polε or POLE1-POLE2 only. Results are reported as mean +/- SD.

suppression of ISC biosynthesis in Basal-like breast cancer, by loss of NFS1, leads to increased levels of DNA damage and loss of viability and identified Polε as the ISC-containing protein that underlies this phenotype[20]. Our data further solidify this hypothesis and suggests that dysfunctional Polε activity might more broadly underlie genetic instability associated with iron metabolism dysfunctions.

In addition to iron metabolism genes, components of the CHTF18-RFC alternative clamp loader complex emerged as the most critical and conserved factors required for viability of POLE3-POLE4 KO cells. Consistently with a *bone fide* synthetic lethal interaction we were not able to generate CHTF18-POLE3/POLE4 double KO cells and transient silencing of POLE3-POLE4 led to a rapid block of proliferation of CHTF18 KO cells. A similar effect was observed in POLE3 and POLE4 KO cells upon silencing of all the specific components of the CHTF18-RFC complex. This phenomenon was associated with a marked reduction of fork extension rates, as measured by DNA fiber assay, and increased fork stalling events pointing to inefficient DNA synthesis and extensive fork stalling, incompatible with cell viability. Thus, CHTF18-RFC and POLE3-POLE4 represent two essential in vivo tiers of regulation of Polε function and genome stability.

The mechanism by which the CHTF18-RFC complex is able to open and load PCNA has been recently elucidated by He et al.[24]; comparison with structures of RFC1-5 suggested the highly motile nature of CHTF18 is the reason why the CHTF18-RFC2/5 complex is less efficient than the canonical clamp loader at loading PCNA; this explains why CHTF18-RFC is not able to load PCNA on the lagging strand. Subsequent work showing how budding yeast CTF18-RFC can displace Polε from the shared DNA substrate, the primer template junction, to load PCNA hinted at why RFC1-5 is unable to function on the leading strand[23]. Further work, using yeast proteins, demonstrated these mechanisms in the context of DNA replication using a reconstituted CMG[28]. Taken together this provides an explanation of why specialised PCNA loaders are required within the eukaryotic replisome and explains mechanistically, how CHTF18-RFC might promote leading strand synthesis. However, if and how human cells might survive in the absence of CHTF18-RFC remained unknown. Here we showed that, while CHTF18 is dispensable for viability, genetic deletion or transient silencing of the accessory subunits of Polε, POLE3-POLE4, leads to a complete loss of viability, denoting a profound synthetic lethal effect.

We and others have previously showed that POLE3-POLE4 are required to maintain Polε stability and leading strand H3-H4 redeposition[5–8]. However where POLE3-POLE4 are located within the full Polε complex and if and how they contribute to Polε enzymatic activities remained unknown. Previous work in budding yeast has shown that the small accessory subunits, Dpb3-Dpb4, sit between the two domains of the catalytic Pol2 subunit to confer a rigid conformation to the Polε complex[16]. Alongside interactions with both the catalytic and non-catalytic domain of Pol2, the interaction between Dpb3-Dpb4 and Pol2 is mediated by a mooring helix located in the flexible linker between the two domains of Pol2. However, inconsistently with

the existence of this rigid conformation, structures of the CMG-E complex at replication forks could only resolve POLE2, which makes direct contact with the CMG, and the non-catalytic domains of POLE1[12,13]. The inability to locate the catalytic domain of POLE1 and POLE3-POLE4 pointed to a high level of flexibility of these domains. A subsequent AlphaFold2 modelling study suggested that further interactions with the CHTF18-RFC complex, not included in the original reconstitution, may anchor the catalytic domain in place[29]. More recent structures of CMG-E including the CHTF18-RFC complex have been able to show the existence of at least two distinct conformations for the catalytic domain of Polε[28]; however, neither of these conformations represent the rigid confirmation observed for the yeast complex. The high degree of flexibility is likely required to enable the polymerase to position itself where the ssDNA emerges from the CMG helicase and also allow handover of the DNA substrate from the polymerase to the clamp loader when required. Recent structures of Polε in complex with PCNA, have revealed the extensive interface between the polymerase and the DNA clamp providing the basis of processive leading strand DNA synthesis in the presence of PCNA[14,15] However, similarly to the flexibility observed in the CMG structures, these resulted in resolving only the catalytic domain of POLE1 in concert with PCNA and a DNA substrate, with the non-catalytic subunits of the polymerase unresolved despite inclusion of the full polymerase complex.

Our AlphaFold3 model, validated by cross linking mass spectrometry and immunoprecipitation experiments in human cells, suggest that POLE3-POLE4 interact with POLE1 via a conserved mooring helix, similarly to what previously shown for its budding yeast orthologue. However, there is lack of evidence supporting the existence of a rigid conformation and the presence of additional crosslinks with the flanking regions of POLE1 instead suggests a high level of flexibility of the complex. We were also able to observe that in vitro loss of POLE3-POLE4 resulted in reduced rates of primer extension and exonucleolytic activity. This is in agreement with previous work in budding yeast that suggested a role for Dpb3-Dpb4 in promoting processivity of full Polε[25]. Our biochemical reconstitution suggests that dsDNA binding by POLE3-POLE4 is crucially responsible for these phenotypes. Consistently with this, further AlphaFold3 modelling of Polε in the presence of a leading strand substrate showed that the anchoring of the accessory subunits to the flexible linker region allowed them to make multiple non-specific interactions between POLE3-POLE4 and newly synthesised dsDNA. It is possible that this confirmation was not observed in the recent Polε-PCNA cryoEM structures due to an insufficiently long DNA substrate being provided. Mutation of critical residues in POLE4 significantly affected both dsDNA binding and extension of short substrates in vitro, while similarly impacting the exonuclease activity of the enzyme.

To explore the interplay between POLE3-POLE4 and CHTF18-RFC we also generated an AlphaFold3 model of Polε bound to both PCNA and DNA. The model suggests that in the presence of PCNA the

remaining flexibility of the linker still enables the POLE3-POLE4 sub-units to reach the nascent dsDNA protruding from behind the clamp, increasing the affinity between the polymerase and its substrate. We speculate that this flexibility may also be important for the other roles of POLE3-POLE4 at the replication fork such as histone chaperoning[5-7]. Consistently with a cooperative model of DNA binding and substrate stabilization, loss of POLE3-POLE4 impacted DNA synthesis in radio-active extension assays of long substrates even in the presence of CHTF18-RFC loaded PCNA.

Inter-origin distances in human cells can be 100 Kb or more. Polε, as leading strand Polymerase, has to sustain uninterrupted DNA synthesis for potentially more than 50 Kb without detaching from its substrate. We speculate this double-tier mechanism is crucial to explain the exceptional processivity of this enzyme.

In summary our work defined the genetic and biochemical basis of processive human leading strand synthesis and the consequences of loss of its regulation in genome stability and cellular homeostasis.

## Methods

### Cell lines and culture
The human HeLa TRex Flip In (HTF) original cell lines were obtained from the Boulton lab (Francis Crick Institute) while 293 T cells were provided by Zuzana Horejsi. RPE1 p53-/- WT, POLE3 KO and POLE4 KO were obtained from Daniel Durocher. Parental HTF and RPE1 p53-/- cell lines and their derivative KO for CHTF18, POLE3 and POLE4 were maintained in DMEM (Invitrogen) supplemented with 10% FBS and 1% penicillin-streptomycin.

### Lentivirus production
$20 \times 10^6$ HEK293T cells were seeded into a T175 flask the day before transfection in order to reach ~90% confluency on the next day. On the day of transfection, media was replaced with 18 ml of warm Opti-MEM (ThermoFisher) and cells were incubated at 37 °C for 30 min-utes. Transfection mix 1 contained 3 ml of Opti-MEM plus 30 μg of transfer vector, 25 μg of psPax2 (Addgene #12260) and 10 μg of pMD2.G (Addgene #12259). 95 μl of P3000 reagent (ThermoFisher) was added before incubating at room temperature for 5 minutes. Transfection mix 2 contained 3 ml of Opti-MEM plus 95 μl of Lipo-fectamine 3000 (ThermoFisher) and was incubated at room tem-perature for 5 minutes as well. Transfection mixes 1 and 2 were then combined (1:1 ratio) and incubated at room temperature for 10 minutes to form DNA-lipid complexes. 6 ml of Opti-MEM media from the T175 flask were removed and replaced with 6 ml of DNA-lipid complex and incubated at 37 °C. After 6 hours on HEK293T cells, media containing the transfection reagents was removed and replaced with 25 ml complete DMEM. ~55 h after transfection, supernatant containing lentiviral particles was harvested and filtered through a 0.45 μm filter. Cleared viral supernatant was then aliquoted and stored at −80 °C before use.

### Genome-wide CRISPR screens
HTF WT-Cas9, HTF POLE4 KO-Cas9, RPE1 p53-/- WT-Cas9 and RPE1 p53-/- POLE4 KO-Cas9 cell lines were transduced with VBC_top3 gRNA library[30] in the presence of 8 μg/ml polybrene. Virus containing media was replaced with fresh media after 24 hours. Three days post trans-duction, the percentage of BFP+ cells was determined by flow cyto-metry. In all cases a transduction rate of ~30% was reached which corresponds to a coverage of >350 cells per gRNA, which was main-tained throughout the screen. Puromycin selection (1,5 μg/mL for HTF and 30 μg/mL for RPE1 p53-/-) was initiated and maintained for four days until %BFP was >90%. Baseline cell pellets were collected and cells were split into two technical replicates for screen maintenance. During the screen, cells were split every 3/4 days and cell pellets were col-lected at midpoint (day 6/7) and endpoint (day 13/14).

### Genomic DNA isolation and Next Generation Sequencing (NGS)
Genomic DNA (gDNA) was isolated using the QIAamp DNA Blood Maxi Kit (Qiagen) according to manufacturer's instruction. NGS libraries were prepared in a two-step process. First, the integrated lentiviral cassette containing the gRNA was amplified from gDNA. The Forward primer 5′ ACACTCTTTCCCTACACGACGCTCTTCCGATCTCTTGTGG AAAGGACGAAACA 3′ and the reverse primer 5′ GTGACTGGAGTTCA-GACGTGTGCTCTTCCGATCTACCCAGACTGCTCATCGTC 3′ were used for this. PCR reactions with 5 μg gDNA per well were set up using the Q5 Hot Start High-Fidelity 2× Master Mix (NEB #M0494) in a total volume of 50 μl. PCR reactions were scaled accordingly to amplify the gRNAs at a coverage of at least 200-fold. The PCR products were then pooled in each group and purified using QIAquick PCR Purification Kit (Qiagen #28106). Final NGS libraries were generated using 2.5 ng of the purified 1st PCR product using the dual-indexing Illumina-compatible DNA HT Dual Index kit (Takara #R400661). 2nd PCR products were purified with AMPure XP beads (Beckman Coulter #A63881) at an 0.7 ratio. Purified 2nd step libraries were quantified using the Qubit dsDNA Quantification Assay Kit (ThermoFisher) and sequenced on NovaSeq6000 by PE50bp with a 30% PhiX spike-in.

### CRISPR screen analysis
For each CRISPR pooled screen, DNA sequencing data was generated by the NovaSeq6000 platform and FASTQ files were produced using bcl2fastq2. The reads were mapped to the VBC_top3 gRNA reference library and exact matches were counted using AstraZeneca's in-house CRISPR counting algorithm (unpublished) giving a gRNA count matrix. Count data were processed through MAGeCK version 0.5.9.5 using its Test function with knock-out cell samples as 'test' samples and parental WT samples as 'controls'[31]. For each knock-out versus parental WT contrast, gene-level log2 fold-changes and adjusted p-values were extracted and compared through timepoints and cell lines.

### Generation of CRISPR knockout cell lines
The Lenti-CRISPR V2-Puro plasmid targeting CHTF18 was purchased from GenScript. HTF were seeded in 10 cm dishes and transfected 24 h later with the Lenti-CRISPR V2-Puro plasmid using Lipofectamine 2000. 48 hours after transfection cells were selected in media con-taining 1 μg/ml puromycin. Cells were then seeded at limiting dilution in 96 well plates and grown for 7–10 days to allow single colonies to form. Single clones were subsequently isolated, expanded and screened for loss of CHTF18 by Western blotting.

### Western blot analysis of cell lysates
Cells were lysed in a buffer containing 50 mM HEPES pH 7.5, 1% (vol/vol) Triton X-100, 150 mM NaCl and 5 mM EGTA, with Protease and Phosphatase inhibitors (ROCHE). Lysates were clarified by centrifuga-tion (16000 x g 30 min at 4 °C) and protein concentration was esti-mated by BRADFORD assay (SIGMA). Equal amounts of proteins were loaded on NuPAGE 4–12% Bis-Tris gels and transferred onto nitro-cellulose membrane (Amersham). Membranes were blocked in 5% milk in PBST (PBS-Tween 0.1%) and incubated with primary antibodies and HRP-conjugated secondary antibodies.

### Flag Immunoprecipitations
Cells were transiently transfected with Flag-tagged constructs using Lipofectamine 2000. Protein extracts were prepared in lysis buffer as described in western blot methods. For Immunoprecipitation experi-ments, 2 mg of the whole-cell extract was incubated for 2 h at 4 °C with 20 μl of M2 anti-FLAG agarose (Sigma-Aldrich). Beads were pelleted and washed three times in 20x bead volume of the lysis buffer. Bound proteins were eluted by boiling in 2x LSB (Laemmli sample buffer) for 5 min. Inputs represent 5% of the extracts used for IP.

### siRNA transfections

Cells were grown to 40–50 % confluency and transfected with 20 nM siRNAs against human NCOA4, IRP2, POLE3, POLE4, CHTF18, CHTF8 and DSCC1 or a negative control (ON TARGETplus SMARTpool, Dharmacon) using Lipofectamine RNAiMAX (Thermo Fisher) according to manufacturer instructions. siRNAs and Lipofectamine were initially diluted in Opti-MEM Medium and mixed after 5 min incubation. After an additional 15 min, the transfection mix was directly added to the cells. 24 hours later, cells were seeded for CFA assays or western blot (pellets were collected in this case at 48 hours).

### IncuCyte Live cell imaging

To investigate cell proliferation, cells were transfected with the indicated siRNAs and seeded, after 24 h, at 1000 cells/well in 96-well plates. Cells were left to adhere in the incubator for additional ~18 hrs. Plates were incubated in an IncuCyte® S3 Live-Cell Analysis System (Sartorius) at 37 °C for 120 h. The first image was captured 3 h after inserting plate into the equipment and the 10 X objective captured 4 fields of view every 3 h. Cell confluency was calculated by image-based measurements of cell growth using the IncuCyte Analysis Software Version 2023 A (Sartorius).

### Colony forming assays

For clonogenic survival assays, 500 cells were seeded per well in 6 well plate format in technical triplicate. Cells were then grown for a further 7–10 days. Surviving colonies were stained using crystal violet, imaged and stained area was analysed using imageJ. Cells were normalised to siRNA CTR transfected or untreated samples.

### DNA Fiber assay

DNA fiber assay was essentially performed as described in Bellelli et al., 2018[7]. Briefly, cells of the indicated genotypes were pulse labelled with 20 µM CldU for 20 min and subsequently pulse labelled with 200 µM IdU for 20 min. Cells were trypsinized, washed in PBS, counted and resuspended at a concentration of $5 \times 10^5$ in PBS. 2.5 µl of cell suspension were spotted on clean glass slides and lysed with 7.5 µl of 0.5% SDS in 200 mM Tris-HCL, pH 7.4, 50 mM EDTA (10 min, R.T.). Slides were tilted (15° to horizontal), allowing a stream of DNA to run slowly down the slide, air dried and then fixed in methanol/acetic acid (3:1) for 15 min at R.T. Acid-treated slides (45 min R.T.) were blocked in 1% BSA/PBS for 45 min at R.T. and incubated with rat anti-BrdU monoclonal antibody (1:1,000 over night; Abcam) and mouse anti-BrdU monoclonal antibody (1:500 1 h R.T.; Becton Dickinson). After 3 washes in PBS, slides were incubated with a mixture of Alexa Fluor 488 rabbit anti-mouse and Alexa Fluor 594 goat anti-rat antibodies (1:500 R.T.; Invitrogen) for 45 min at room temperature and mounted in PBS/Glycerol 1:1. Fibers images were acquired with a Nikon Eclipse Ti-E inverted Confocal Microscope using 63× objective lens and quantification was performed using ImageJ.

### Cell Cycle Analysis

Cell cycle analysis was performed as previously described[7]. Briefly, cells were harvested and fixed in 70% ice-cold methanol. After washing in PBS 1X, cells were resuspended in a buffer containing 0.1% NP40, 20 µg/mL propidium iodide (PI) and 100 mg/mL RNase A for at least 30 minutes in the dark. Data were collected with a BD LSR Fortessa and analysed using FlowJo V.10 software.

### EdU incorporation and γH2AX detection with flow cytometry

EdU (5-ethynyl-2′-deoxyuridine), supplied with the Click-iT™ EdU Alexa Fluor™ 647 Flow Cytometry Assay Kit (#C10424, Thermo-Fisher), was diluted in DMSO to a stock concentration of 10 mM and stored at −20 °C. Cells were incubated with 10 µM EdU for 1 h and then harvested and resuspended in media at a final concentration of ~$5 \times 10^5$ cells/mL. Cell suspension was centrifuged at 500 xg for 5 min

and the supernatant was discarded. Cell fixation, permeabilization and Click-iT™ reaction were performed according to manufacturer instructions in the Click-iT™ EdU Alexa Fluor™ 647 Flow Cytometry Assay Kit (#C10424, ThermoFisher). Briefly, cells were resuspended in Click-iT™ fixative for 15 mins at RT and then washed in 1% BSA/PBS. The cell pellet was resuspended in saponin-based permeabilization and wash reagent buffer before adding the click-iT™ reaction cocktail. After 30 mins at RT, cells were centrifuged and the cell pellets were resuspended and incubated in 100 µL anti-γH2AX (1/100 in 1% BSA/PBS) for 1 h at RT. Cells were then washed in saponin-based permeabilization and wash reagent buffers and the supernatant was discarded. Cell pellets were subsequently resuspended and incubated in 100 µL Alexa Fluor™ 488 anti-Rabbit IgG (1/200 in 1 % BSA/PBS) for 1 h at RT. Finally, cells were washed and resuspended in 250 µL of 1 µg/mL DAPI (10236276001, Sigma-Aldrich) in saponin-based permeabilization and wash reagent. Cells were left to incubate overnight at 4 °C prior to flow cytometer analysis. Data were collected with a BD LSR Fortessa and analyzed using FlowJo V.10 software.

### RNA extraction and quantitative PCR

RNA was extracted using the RNeasy®mini kit (74106, Qiagen) according to the manufacturer's instructions and quantified with a nanodrop spectrophotometer (ThermoFisher). 1 µg RNA was used for cDNA synthesis using the QuantiTect® Reverse Transcription kit (205313, Qiagen). Quantitative PCRs (qPCRs) were performed using a QuantStudio™ 7 Flex (ThermoFisher) and PowerUp™ SYBR™ Green Master Mix. Thermocycling was performed as follows: 50 °C for 2 min, 95 °C for 2 min then 40 cycles of 95 °C for 1 sec and 60 °C for 30 sec. GAPDH and RPS18 were used as reference genes to normalise expression levels using the $2^{-\Delta\Delta CT}$ method. The sequences of qPCR primers were designed by using the web interface Primer3Plus.

### Crosslinking Mass Spectrometry

Crosslinked complexes were reduced with 10 mM dithiothreitol and alkylated with 50 mM iodoacetamide. They were then digested with trypsin at an enzyme-to-substrate ratio of 1:100, for 1 h at RT and further digested overnight at 37 °C following addition of trypsin at a ratio of 1:20. The peptide digests were then fractionated batch-wise by high pH reverse phase chromatography on micro spin TARGA C18 columns (Nest Group Inc, USA), into five fractions (10 mM $NH_4HCO_3$/10% (v/v) acetonitrile pH 8.0, 10 mM NH4HCO3/20% (v/v) acetonitrile pH 8.0, 10 mM $NH_4HCO_3$/30% (v/v) acetonitrile pH 8.0, 10 mM $NH_4HCO_3$/40% (v/v) acetonitrile pH 8.0 and 10 mM $NH_4HCO_3$/80% (v/v) acetonitrile pH 8.0). The fractions (120 µL) were evaporated to dryness in a CentriVap concentrator (Labconco, USA) prior to analysis by LC-MS/MS.

Lyophilized peptides were resuspended in 1% (v/v) formic acid and 2% (v/v) acetonitrile and analysed by nano-scale capillary LC-MS/MS using a Vanquish Neo UPLC (ThermoScientific Dionex, USA) to deliver a flow of approximately 250 nL/min. A PepMap Neo C18 5 µm, 300 µm x 5 mm nanoViper (ThermoScientific Dionex, USA) trapped the peptides before separation on a 50 cm EASY-Spray column (50 cm × 75 µm ID, PepMap C18, 2 µm particles, 100 Å pore size: Thermo-Scientific, USA). Peptides were eluted with a gradient of acetonitrile over 90 minutes. The analytical column outlet was directly interfaced via a nano-flow electrospray ionisation source, with a quadrupole Orbitrap mass spectrometer (Orbitrap Exploris 480, ThermoScientific, USA). MS data were acquired in data-dependent mode using a top 10 method, where ions with a precursor charge state of 1+ and 2+ were excluded. High-resolution full scans (R = 60000, m/z 380-1800) were recorded in the Orbitrap followed by higher energy collision dissociation (HCD) (stepped collision energy 30 and 32% Normalized Collision Energy) of the 10 most intense MS peaks. The fragment ion spectra were acquired at a resolution of 15,000 and a dynamic

exclusion window of 20 secs was applied. For data analysis, Xcalibur raw files were converted into the MGF format using Proteome Discoverer version 2.3 (ThermoScientific) and used directly as input files for MeroX[76]. Searches were performed against an ad hoc protein database containing the sequences of the proteins in the complex and a set of randomized decoy sequences generated by the software. The following parameters were set for the searches: maximum number of missed cleavages: 3; targeted residues K, S, Y and T; minimum peptide length 5 amino acids; variable modifications: carbamidomethylation of cysteine (mass shift 57.02146 Da), methionine oxidation (mass shift 15.99491 Da); DSBU modified fragments: 85.05276 Da and 111.03203 Da (precision: 5 ppm MS and 10 ppm MS/MS); False Discovery Rate cut-off: 5%. Finally, each fragmentation spectrum was manually inspected and validated.

### Protein expression and purification

The sequences codifying for WT POLE1, POLE2, POLE3 and POLE4 were originally obtained in the pFAST BAC1 vector from Dr Tahir Tahirov. A streptavidin tag was included at the N-terminal region of POLE1 by PCR-methods and the coding sequences were cloned into a single vector using the biGBac system[32]. POLE1-4, POLE1-2, POLE1-4 DB mutant (POLE4_A44L_R45D_K47E_K51E) and POLE1-4 ISC mutant (POLE1_C651S_C654S) bacmids were then transfected into Sf9 cells to produce baculoviruses. After 2 cycles of virus amplifications, P2 were used to infect fresh Sf9 cells in large scale (1-2 liters) for protein expression and purification. Cell pellets were re-suspended in lysis buffer containing 25 mM Hepes pH 7.5, 200 mM NaOAc, 0.5 mM TCEP, 5% glycerol supplemented with 30 U DNASE Turbo (Thermo Fisher Scientific) and cOmplete, EDTA-free Protease Inhibitor Cocktail (Merck). The cells were then disrupted by sonication, and the resulting lysate was clarified by centrifugation at 38758 ×g for 60 minutes at 4 °C. The supernatants were applied to a 5 ml HiTrap TALON crude column (Cytiva) washed first with lysis buffer, followed by lysis buffer supplemented with 4.5 mM imidazole and then eluted using the same buffer supplemented with 300 mM imidazole. The eluted proteins were diluted 1:1 with lysis buffer before application to a 1 ml StrepTrap XT Chromatography Column (Cytiva), washed with lysis buffer and eluted with 250 mM Hepes pH 7.5, 200 mM NaOAc, 0.5 mM TCEP, 5% glycerol supplemented with 50 mM Biotin (iba). Following concentration, the samples were applied to Superdex200 10/300 size exclusion column (GE Healthcare) equilibrated with 10 mM Hepes pH 7.5, 150 mM NaCl, 0.5 mM TCEP, 5% glycerol, 0.002% TWEEN 20. Proteins were eluted in the same buffer and then diluted 5-fold with 25 mM Hepes pH 7.5, 0.5 mM TCEP, 0.005% NP-40, 5% glycerol, 500 mM NaOAc, followed by concentration. A final concentration of 20% glycerol was added before snap-freezing the proteins in liquid nitrogen.

CHTF18-RFC2_5-DSCC1-CHTF8 was produced by transfecting Sf9 cells with baculoviruses coding for constructs containing STREP-CHTF18, RFC2, RFC3, RFC4, HIS-RFC5, DSCC1 and CHTF8 produced using the biGBac system[28]. After 2 cycles of virus amplifications, P2 were used to infect fresh Sf9 cells in large scale (1-2 liters) for protein expression and purification. The cell pellet was re-suspended in lysis buffer containing 25 mM Hepes pH 7.5, 200 mM NaCl, 0.5 mM TCEP, supplemented with 30 U DNASE Turbo (Thermo Fisher Scientific) and cOmplete, EDTA-free Protease Inhibitor Cocktail (Merck). The cells were then disrupted by sonication, and the resulting lysate was clarified by centrifugation at 38758 ×g for 60 minutes at 4 °C. The supernatants were applied to a 5 ml HiTrap TALON crude column (Cytiva) washed first with lysis buffer, followed by lysis buffer supplemented with 4.5 mM imidazole and then eluted using the same buffer supplemented with 300 mM imidazole. The eluted protein was diluted 1:1 with lysis buffer before application to a 1 ml StrepTrap XT Chromatography Column (Cytiva), washed with lysis buffer and eluted with 250 mM Hepes pH 7.5, 200 mM NaCl, 0.5 mM TCEP, supplemented

with 50 mM Biotin (iba). Following concentration, the samples were applied to Superdex200 3.2/300 equilibrated with 150 mM NaCl, 0.5 mM TCEP, 0.005% Poly20, 25 mM Hepes pH 7.5, 10% glycerol. Peak fractions were pooled, concentrated and snap-freezed in liquid nitrogen.

RPA was expressed in Bl21 (DE3) Rosetta cells. The cells were grown at 37 °C in LB + 100 μg/ml ampicillin to an $OD_{600}$ of 0.6. Expression was induced by addition of 0.5 mM IPTG and the culture was grown overnight at 20 °C before cell pellets were collected by centrifugation at 6238 xg for 10 minutes at 4 °C and stored at −80 °C. Cell pellet was resuspended in lysis buffer 25 mM Hepes pH 7.5, 200 mM NaCl, 0.5 mM tris(2-carboxyethyl)phosphine (TCEP), supplemented with 30 U DNASE Turbo (Thermo Fisher Scientific) and cOmplete, EDTA-free Protease Inhibitor Cocktail (Merck). The cells were then disrupted by sonication, and the resulting lysate was clarified by centrifugation at 38758 × g for 60 minutes at 4 °C. The supernatants were applied to a 5 ml HiTrap TALON crude column (Cytiva) washed first with lysis buffer, followed by lysis buffer supplemented with 4.5 mM imidazole and then eluted using the same buffer supplemented with 300 mM imidazole. The eluted proteins were diluted 1:4 with 25 mM Hepes pH 7.5, 0.5 mM TCEP, 5% (v/v) glycerol before application to a 5 ml HiTrap Q HP anion exchange chromatography column (Cytiva) equilibrated with 25 mM Hepes pH 7.5, 0.5 mM TCEP, 5% (v/v) glycerol. Protein was eluted with a 20 CV gradient from 50 mM to 1000 mM NaCl. Following concentration, the samples were applied to Superdex200 10/300 size exclusion column (GE Healthcare) equilibrated with 25 mM Hepes pH 7.5, 300 mM NaCl, 0.5 mM TCEPT, 5% Glycerol. Peak fractions were pooled, concentrated and snap-freezed in liquid nitrogen.

PCNA was expressed in Bl21 (DE3) Rosetta cells. The cells were grown at 37 °C in LB + 100 μg/ml ampicillin to an OD600 of 0.6. Expression was induced by addition of 0.5 mM IPTG and the culture was grown overnight at 20 °C before cell pellets were collected by centrifugation at 6238 x g for 10 minutes at 4 °C and stored at −80 °C. Cell pellet was resuspended in lysis buffer 25 mM Hepes pH 7.5, 200 mM NaCl, 0.5 mM tris(2-carboxyethyl)phosphine (TCEP), supplemented with 30 U DNASE Turbo (Thermo Fisher Scientific) and cOmplete, EDTA-free Protease Inhibitor Cocktail (Merck). The cells were then disrupted by sonication, and the resulting lysate was clarified by centrifugation at 38758 × g for 60 minutes at 4 °C. The supernatants were applied to a 5 ml HiTrap TALON crude column (Cytiva) washed first with lysis buffer, followed by lysis buffer supplemented with 4.5 mM imidazole and then eluted using the same buffer supplemented with 300 mM imidazole. The eluted proteins were diluted 1:4 with 25 mM Hepes pH 7.5, 0.5 mM TCEP, 5% (v/v) glycerol before application to a 5 ml HiTrap Q HP anion exchange chromatography column (Cytiva) equilibrated with 25 mM Hepes pH 7.5, 0.5 mM TCEP, 5% (v/v) glycerol. Protein was eluted with a 20 CV gradient from 50 mM to 1000 mM NaCl. Following concentration, the samples were applied to Superdex200 10/300 size exclusion column (GE Healthcare) equilibrated with 25 mM Hepes pH 7.5, 300 mM NaCl, 0.5 mM TCEPT, 5% Glycerol. Peak fractions were pooled, concentrated and snap-freezed in liquid nitrogen.

### DNA substrates

Purified oligonucleotides were purchased from Sigma-Aldrich. The DNA sequence 5′-[6FAM]-AGCTACCATGCCTGCACGAATTAAGCAATT CGTAATCATGGTCATAGCT-3′ was used for EMSA ssDNA.

For EMSA dsDNA, the following DNA sequences were used: 5′-[6FAM]-AGCTACCATGCCTGCACGAATTAAGC AATTCGTAATCATGGT CATAGCT-3′ and 5′-AGCTATGACCATGATTACGAATTGCTTAATTCGT GCAGGCATGGTAGCT-3′. The two oligonucleotides were mixed in a 1:1.25 molar ratio in annealing buffer (50 mM Tris-HCl, pH 7.5, 100 mM NaCl, 10 mM $MgCl_2$), heated at 95 °C for 5 minutes, and cooled down for 4 hours.

For the EMSA Fork DNA substrate, the following DNA sequences were used: 5'-[6FAM]-AGCTACCATGCCTGCACGAATTAAGCAATTCG TAATCATGGTCATAGCT-3', 5'-CTACAGTTCGTCAGGATTCC-3', and 5'-AGCTATGACCATGATTACGAATTGCTTGGAATCCTGACGAACTGTA G-3'. The three oligonucleotides were mixed in a 1:1.4:1.25 molar ratio, respectively, in annealing buffer (50 mM Tris-HCl, pH 7.5, 100 mM NaCl, 10 mM MgCl$_2$), heated at 95 °C for 5 minutes, and cooled down for 4 hours.

For the Primer Extension assay and Exonuclease assay, the following DNA sequences were used: Primer (50-mer): 5'-[6FAM]-GAT-CAGACTGTCCTTAGAGGATACTCGCTCGCAGCCGTCCACTCAACTCA-3' and Template (80-mer): 5'-CAGCTTGATAGTCAGTGACGTTGTTC TGGATGAGTTGAGTGGACGGCTGCGAGCGAGTATCCTCTAAGGACAG TCTGATC-3'. The two oligonucleotides were mixed in a 1:1.2 molar ratio in annealing buffer (50 mM Tris-HCl, pH 7.5, 100 mM NaCl), heated at 95 °C for 5 minutes, and cooled down for 4 hours.

## Electrophoretic Mobility Shift Assay (EMSA)

Purified proteins were diluted in 25 mM Hepes pH 7.5, 100 mM NaCl, 10% glycerol, 0.5 mM EDTA, 1 mM DTT, 0.01% NP-40 and mixed with specific DNA substrates in 5x reaction buffer (175 mM Hepes pH 7.4, 100 mM NaCl, 10 mM DTT, 500 ug/ml BSA). Each reaction contains 30 nM DNA mixed with different protein concentrations (30 nM, 60 nM, 90 nM, 120 nM). The samples were incubated on ice for 20 minutes to allow formation of protein-DNA complexes before being loaded onto the gel. Samples were run on Novex™ TBE Gels, 4–12% (Invitrogen) for 1 hour at 100 V at 4 °C. Gels were visualised using BIO-RAD ChemiDoc MP Imaging System. EMSA gels were quantified by using ImageJ software version 1.5t (ImageJ) and data were plot by using the "Specific binding with Hill slope" function in GraphPad Prism version 10.4.1627 (https://www.graphpad.com/features).

## Primer Extension and Exonuclease Assays

Purified proteins were diluted to 30 nM in 25 mM Hepes pH 7.5, 25 mM NaCl, 2 mM MgCl$_2$, 5% glycerol, 0.01% Triton, 1 mM DTT and mixed with 30 nM primer-template (50/80-mer) in 25 mM Hepes pH 7.5, 25 mM NaCl, 2 mM MgCl$_2$, 5% glycerol, 0.01% Triton, 1 mM DTT, 0.1 mg/ml BSA, 0.1 mM dNTPs. For the exonuclease assay, dNTPs were not added to the reaction. Reactions were performed at 30 °C for different time points (0 s, 10 s, 20 s, 40 s, 80 s, 120 s for the Primer Extension Assay and 0 s, 2 min, 5 min, 10 min for the Exonuclease assay). Reactions were stopped by adding 95% formamide, 20 mM EDTA, 0.1% bromophenol blue, followed by boiling at 95 °C for 10 minutes. Samples were run on Invitrogen™ Novex™ TBE-Urea Gels, 10% for 45 minutes at 185 V using 1X TBE running buffer pre-warmed at 65 °C. Gels were visualised using BIO-RAD ChemiDoc MP Imaging System and quantified by using ImageJ software version 1.5t (ImageJ). Data were plot by using the "exponential one phase decay" function in GraphPad Prism version 10.4.1627 (https://www.graphpad.com/features).

## Radioactive extension assay

The Oligonucleotide (500 nM; sequence: 5' GAATAATGGAAGGGTTA-GAACCTACCAT) was annealed to 50 nM M13mp18 single-stranded DNA (New England Biolabs) in a buffer containing 10 mM Tris-HCl (pH 7.5), 100 mM NaCl, and 5 mM EDTA. The mixture was heated to 95 °C and then gradually cooled to room temperature. The reaction was carried out at 37 °C in a buffer containing 25 mM Hepes (pH 7.5), 100 mM potassium glutamate, 0.01% NP-40-S, 1 mM DTT, 10 mM Mg(OAc)$_2$, 0.1 mg/ml BSA, 5 mM ATP, 200 μM CTP, GTP, UTP, 30 μM dATP, dCTP, dGTP, dTTP, and 33 nM α-[$^{32}$P]-dCTP. The primed template (2 nM) was first incubated with 400 nM RPA, 20 nM PCNA, and 20 nM CHTF18-RFC2_5-DSCC1-CHTF8 for 10 minutes at room temperature. The reaction was then initiated by adding 20 nM Polε

or POLE1-2. At the indicated time points, reactions were stopped by adding EDTA to a final concentration of 50 mM. Unincorporated α-[$^{32}$P]-dCTP was removed using illustra MicroSpin G-50 columns (GE Healthcare). Reactions were run on a 0.8% alkaline agarose gel in 30 mM NaOH and 2 mM EDTA for 16 hours at 24 V. Gels were then fixed with 5% cold trichloroacetic acid and dried for 90 minutes in a hybridization oven. A phosphor screen was then applied for 3 h on the dried gel in a dark room. After 3 h, the phosphor screen was scanned using Cyclone phosphor detector (Packard Biosciences, UK).

## AlphaFold 3 modelling

Sequences were taken from Uniprot accession codes POLE1:Q07864, POLE2:P56282, POLE3:Q9NRF9, POLE4:Q9NR33 and PCNA:P12004, with the following DNA sequences used to provide a extending DNA substrate structure 5'-ACAACCAAACCCAAAACCCCAAAAACCCCCAAA AAACCCCCAAAAAAACCCCCCAAAAAAAACCCCCCCCAAAAAAAA ACCCCCCCCC-3' and 5'TTTTTTTTGGGGGGGTTTTTTTGGGGGGTT TTTTGGGGGTTTTTGGGGTTTTGGGTTTGGTTGT-3'. Jobs were submitted to the AlphaFold3 server[33] and PAE viewer server was used to produce and assess the resulting PAE plots for all models[34]. All structure figures were produced using PyMOL (v 2.2.2, Schrodinger LLC).

## Quantification and statistical analysis

All statistical analysis were performed using GraphPad Prism 9 software. Statistical details for the experiments are provided in the figure legends. $p < 0.05$ was considered to be significant and classified by asterisks: $p < 0.05$ (*), $p < 0.01$ (**), $p < 0.001$ (***) and $p < 0.0001$ (****).

## Reporting summary

Further information on research design is available in the Nature Portfolio Reporting Summary linked to this article.

## Data availability

The alphafold3 data generated in this study have been deposited in the ModelArchive[35] database under the following accession code ma-ixbas [https://modelarchive.org/doi/10.5452/ma-ixbas] (Human Polymerase Epsilon in complex with DNA and PCNA); ma-c2hux [https://modelarchive.org/doi/10.5452/ma-c2hux] (Human Polymerase Epsilon in complex with DNA) and ma-z8mhq [https://modelarchive.org/doi/10.5452/ma-z8mhq] (Human Polymerase Epsilon). The mass spectrometry proteomics data have been deposited to the ProteomeXchange Consortium via the PRIDE[36] partner repository with the dataset identifier PXD070472. Count data, magic results and library information of the CRISPR screening were uploaded on Zenodo (https://doi.org/10.5281/zenodo.17553423). Source data are provided with this paper.

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

## Acknowledgements

We would like to thank Daniel Durocher for providing RPE1 p53-/- WT, POLE3 KO and POLE4 KO cells, Tahir Tahirov for the original pFastBac-POLE1, -POLE2, -POLE3 and -POLE4 vectors, Joe Yeeles for the pACEBac1 CHTF18, DSCC1 and CHTF8 vectors, Roopesh Anand for help setting up Polε baculovirus production and Dr Charlotte Millership of the Protein Production Facility at Queen Mary University of London for her expert technical support and assistance in this work. Work in R.B. lab is funded by an MRC New Investigator Research Grant (MR/X019098/1) and a Springboard Award from the Academy of Medical Science (SBF008\1109). Work in M.D. lab is funded by Royal Society grant (RG \R2\232017). O. B. is supported by Cancer Research Institute Irvington Postdoctoral Fellowship (CRI14353) and is recipient of the EMBO LT Fellowship (ALTF 617-2022).

## Author contributions

R.B. conceived the project and wrote the manuscript with M.D.; M.D. perform structural modelling and supervised protein expression and purification by A.A.; S.M. and M.J.S. performed and analysed Cross-linking mass spectrometry data; O.B. analysed CRISPR screen data and reviewed the manuscript; M.L, K.K.A., A. K. and D.W. performed and analysed the CRISPR screenings; A.A., M.O. and L.B-.B. performed all the remaining experiments of biochemistry and cell biology.

## Competing interests

A.K., M.L., and D.W. are employees of Cancer Research Horizons. K.K.A. is an employee of AstraZeneca. The remaining authors declare no competing interests.
