## [Transparent Peer Review file · Nature Communications]

The genetic and biochemical basis of human leading strand synthesis

Corresponding Author: Dr roberto Bellelli

Version 0:

Reviewer comments:

Reviewer #1

(Remarks to the Author)

Summary: Using a combination of genetic, cell biology, biochemical and structural modeling approaches, the authors investigated DNA Polymerase ϵ regulation during leading strand synthesis. Their results focused on the role of the CHTF18-RFC complex and the two non-essential Pol ϵ accessory subunits, POLE3 and POLE4.

The authors first mapped genetic factors required to support viability of POLE4 knockout cells but not wild type cells using a whole-genome CRISPR screen. Synthetic lethal interactions were observed between POLE3-POLE4 and iron metabolism genes, as well as the CHTF19-RFC2/5 complex. Effects on proliferation, cell cycle distribution and increased γ H2AX-positive cells were all observed in POLE3 and POLE4 KO cells treated with siNCOA4 (an essential iron regulator). The Iron Sulphur Cluster (ISC) in the catalytic subunit of Pol ϵ was demonstrated to be required for efficient DNA synthesis in vitro, despite being able to engage with its DNA substrate. In order to test the importance of iron regulation on Pol ϵ activity, the authors mutated two conserved cysteines in the Iron Sulfur Cluster of human Pol ϵ and showed that polymerase and exonuclease activities were abolished, despite the fact that the mutant enzyme can still bind DNA substrates.

Components of the CHTF19-RFC2/5 clamp loader were the other major class of genes identified in the genetic screen as required for viability in POLE3-POLE4 KO cells. The authors were unable to generate CHTF18-POLE3-POLE4 double KO cells, suggesting that both are critical for replication, and transient silencing of either POLE3-POLE4 in CHTF18 KO cells or silencing of CHTF18 in POLE3-POLE4 KO cells blocked proliferation and caused a reduction in replication fork extension and increased fork stalling.

The authors then used AlphaFold3 to model the interaction between POLE3-POLE4 and the catalytic POLE1 subunit of Pol ϵ to demonstrate and test, using crosslinking mass spectrometry and IP experiments, a mooring helix that is conserved in yeast. Using EMSAs, The POLE3-POLE4 subunits were shown to be critical for interaction with DS DNA at replication forks, and loss of these subunits resulted in reduced rates of primer extension and exonucleolytic activity.

The main conclusions made by the authors are that there are two modes of regulation of Pol ϵ processivity during leading strand replication. The first involves loading by CHTF18-RFC2/5. The second involves gripping of DS DNA at replication forks by the POLE3-POLE4 subunits of Pol ϵ . As accurate and efficient DNA synthesis and exonucleolytic proofreading by Pol ϵ are vital to prevention of diseases such as cancer, these results provide insight into the multiple layers of regulation of Pol ϵ activity and protein-protein interactions.

The questions being asked are compelling, and the results presented are novel and of interest to those studying DNA replication and how defects in this process contribute to human disease. The methodology used is appropriate to address the questions being asked and the conclusions are consistent with experimental results.

I have only minor suggestions where the results could be strengthened as follows:

1) Lines 126-130 – The statement ‘...pointing to the potential requirement for different replication-coupled DNA repair pathways in these cell lines...’ Can the authors please expand on this statement. I agree that performing this analysis in multiple cell lines was a good approach, and that the two similar cell lines used yielded similar but distinct results. Please expand on what is known regarding DNA repair pathway choice in each of these cell lines and how this may relate to the results presented.

2) Lines 174-176 – ‘... this data suggest that reduced stability of Pole might play at least a partial role in the observed synthetic lethality with iron metabolism factors’. This is a significant result and deserves more attention. The lower level of POLE1 is present in POLE3 and POLE4 KO cells even in cells not treated with the iron chelating agent, DFO, and the levels decrease further during the timecourse of DFO treatment. This reduction in POLE1 level was also shown in the POLE3 and POLE4 KO cells in Figure 4D. Although further investigation would be beyond the scope of this study, the possible

mechanistic reasons for this POLE1 protein level should be discussed. Perhaps this could be explored using structural and/or modeling approaches.

3) Figure 2C – Please alter the colors of the bars for S and G2/M as there is very little contrast between the two, making it difficult to appreciate the changes in % cells.

4) Figure 6D – are the DNA substrates used in the EMSAs the same as those in Figure 3E?

5) Figure 7H and Supplementary Figure S6 – is it possible to quantify and plot the data for extension rate?

Reviewer #2

(Remarks to the Author)

In this study, Agnarelli et al aimed at understanding the function of POLE3 and POLE4, two accessory subunits of leading strand DNA polymerase, Pol epsilon, in DNA replication. First, the authors performed a CRISPR/Cas9 screen and identified genes that when depleted show synthetic lethal phenotypes with POLE4. They identified top candidate genes including those involved in iron metabolism such as NCOA4 and proteins involved in PCNA loading (CHTF18). They validated the effects of NCOA4 and CHTF18 depletion on cells. Furthermore, they mutated iron-sulfur cluster and found that mutations at this cluster affects DNA synthesis in vitro. Based on these results, they concluded that the dependence of POL4 KO cells on NCOA4 are likely due to the dependence of POLE1 on iron-sulfur cluster. It is known that POLE1 interacts with POLE3 and POLE4 through “mooring helix” domain. They mutated the binding interface between POLE1 and POLE4 and identified key residues that when mutated affect these interactions. Furthermore, based on Alpha-fold, they predicted POLE3-POLE4 binds dsDNA and mutations at the putative DNA binding surface affects DNA synthesis in vitro. Based on these results, they conclude that POLE3 and POLE4 are important to work with PCNA to stabilize POL epsilon at leading strand.

I have the following major concerns.

- 1) Fig. 3. NCOA4 has multiple functions. In order to connect synthetic lethality between NCOA4 and POLE4 to iron dependence, the effects of ISC mutant alone and in combination with POLE4 KO on DNA synthesis in cells must be tested.
- 2) Fig. 4. Because CHTF18 and POLE3 and POLE4 function in other processes in addition to DNA replication, the effects of CHTF18 mutants that cannot bind to PCNA on replication forks and cell viability must be tested alone or in combination with POLE3 and POL4 KO.
- 3) Fig. 6. What are the effects of POLE1 W1271R and POLE4 D53K mutation on DNA synthesis in vitro? How do these mutants affect DNA replication forks in cells? Are these mutants also show synthetic growth defects with CHTF18 depletion?
- 4) Fig. 7. Does POLE4 DB mutant affect histone binding? Does this mutant show synthetic defects with CHTF18 depletion?
- 5) Fig. 7H. Please clarify whether PCNA was included in vitro reaction. If not included, the experiments should be repeated by including PCNA in the reaction.

Minor concerns.

- 1) Cite the reference PMID: 30115745 for the role of POLE3 and POLE4 in parental histone transfer. What is evidence of reference 5 showing that POLE3 and POLE4 in parental histone transfer?
- 2) “...survival of POLE3-POLE4 KO cells and, by extension, Pol function”. POLE3 and POLE4 have other functions. Therefore, the statement above is not accurate.

Reviewer #3

(Remarks to the Author)

In this manuscript, Agnarelli et al aim to identify genetic factors involved in leading strand DNA synthesis by DNA polymerase epsilon (Pol-epsilon) in human cells. By performing CRISPR screens in POLE4 knockout cell lines, the authors uncover that both iron metabolism dysfunction and loss of the alternative clamp loader complex CHTF18-RFC sensitise cells. Building upon these findings, the authors assess whether the identified factors are functionally linked to Pol-epsilon by examining their genetic interactions with POLE3-POLE4 and through structural analysis of key residues. Regarding iron metabolism, the authors demonstrate that the combination of iron homeostasis dysregulation and POLE3/POLE4 knockout leads to severe growth defects. With respect to CHTF18-RFC, although its association with the replisome and its role in efficient replication fork progression have been previously reported (Fujisawa et al., 2017, NAR, PMID: 28199690; Baris et al., 2022), the structural and biochemical analyses presented in this study suggest a complementary relationship between CHTF18-RFC and the POLE3-POLE4 complex. Overall, I believe this study provides informative and reasonable evidence that these factors contribute to efficient Pol-epsilon-mediated DNA synthesis. The findings offer valuable insights that merit consideration for publication in Nature Communications. However, in our opinion, the following issues should be addressed prior to publication:

Major issues

1. The authors demonstrate a genetic interaction between iron levels and POLE3-POLE4, resulting in both proliferation defects and elevated γ H2AX levels in G2 phase, which may stem from inefficient replication during S phase. However, I find the evidence linking iron metabolism specifically to leading-strand DNA synthesis to be somewhat indirect. I suggest that a more direct assessment, such as a DNA fiber assay in POLE3/POLE4 knockout cells with and without siNCOA4, could help clarify whether iron metabolism directly impacts leading-strand synthesis.

2. To investigate the potential impact of disrupted iron homeostasis on Pol-epsilon, the authors introduce mutations in amino acids that constitute the iron-sulfur cluster motif of the purified protein. However, it remains unclear whether the observed effects of these mutations are directly attributable to the loss of iron within the protein structure. It is also conceivable that individual mutations may induce broader structural alterations independent of iron coordination. I would encourage the authors to consider an experimental approach that can distinguish between these possibilities.

3. The authors infer that that role of CHTF18 in stimulating efficient synthesis involves its function in PCNA loading. However, evidence supporting this claim is insufficient. This statement could be supported via in vitro experiments showing the dependence of efficient synthesis on PCNA (e.g. by conducting the experiment in Figure 7H +/- PCNA or with PIP-box mutated POLE).

Minor issues

1. Figure 2A/2B. Knockdown of NCOA4 is less efficient in WT compared to POLE3 and POLE4 KO cells. Could residual protein in these cells mean that proliferation is not affected to the same extent?
2. Figure 2G. An untreated control should be provided in this panel, to highlight the differences induced by iron chelation. The data should also be plotted as “change in % confluency” for consistency with the associated legend (line 555) and the other figures.
3. Figure 4A,B,C,D,H,I. Figures are labelled with CTF18, not CHTF18.
4. Figure 7H and Figure S6. Figure is labelled with CTHF18, not CHTF18.
5. Line 69. Correct the spelling of “independent”.
6. Line 69. The wording here is somewhat unclear, without the context of the previous paper. The phrasing can be simplified. Perhaps “these data are independent of Pol-epsilon expression levels and suggest functions for POLE3-POLE4 in controlling DNA replication and genomic stability”.
7. Lines 101-108. The introductory lines of the first section largely repeat information from the introduction. These could therefore be shortened.
8. Line 165. The authors state that redundancy and different expression levels of IRP1 and IRP2 reduces the deleterious effects of depleting IRP2 in RPE1 cells, but no evidence is provided for either redundancy or different expression. If this is speculation, I suggest removing “the” from the sentence.
9. Line 225. These results, combined with the CRISPR screen, do indeed point to a synthetic lethal interaction between POLE3-POLE4 and CHTF18. However, this section would benefit from quantitative details of the CRISPR editing attempted, e.g. how many times the experiment was conducted and how many clones were screened (if they could be isolated).
10. Line 226. The sentence regarding loss of CHTF18 appears to contradict itself by first implying that there is no obvious defects in cell growth but then explaining that the data show a growth defect. This sentence should be refined for clarity of the situation (i.e. that when kept under normal conditions growth appears fine, but defects occur when seeded at low confluency).
11. Line 243. The text directs to “Fig S2”, but it should direct to “Fig. S3”.
12. Line 251. The text states that CHTF18 KO cells present a significant reduction in fork speed, but the significance of this comparison is not displayed in Figure 4H. Please add this comparison to the figure. Also add the same comparison to Figure 4I, which is described as not changing on line 252.
13. Line 276. Please edit this sentence to be more clear about where the revealed cross-links are between. Perhaps “... revealed cross-links between the linker region near the putative mooring helix and both POLE3 and POLE4” is more clear.
14. Line 283. It should be noted that a different cell line is being used for the IP experiments compared to the previous experiments. While we assume that the fundamentals of POLE1 are the same between the cell lines, it is important to make the difference clear in the main text.
15. Line 289. The text states that mutating R1286 in POLE1 affects the interaction with POLE3, but this doesn't seem supported by the IP shown in Figure 5D, where seemingly comparable amounts of POLE3 are pulled down by WT and R1286A. Please either alter the text to reflect this or provide the quantification etc that was used to determine the detrimental effect.
16. Line 345. Please cite the “previous work”.
17. Line 349. Please cite the “recently published work”.
18. Line 373. Again, if a difference in dependencies or processes is only speculative, this should be more clear. Here “the” could be changed to “a” to better reflect this.
19. Line 400. Reference 20 is cited, but it should be 19.
20. Lines 536-542. Since Figure A and B present the same experiment in the same way, the description of B can be shortened to reduce repetition. E.g. “As A, with RPE1...”.
21. Line 544. A description of the significance labels is given, but Figure 2C does not have such labels. Perhaps this text relates to Figure 2E, for which a description of the labels is missing (Line 547). Please fix as appropriate.
22. Line 571. This sentence implies that the substrate and dNTPs are used at equimolar concentrations, rather than the enzyme and substrate. Please reword for clarity. This also appears on line 632 and line 656.
23. Line 601. Please add an explanation of the significance labels for Figure 4E.
24. Line 605. To enhance clarity of the experiment conducted, please replace “replication forks at newly activated replication origins” with “replication forks either side of newly activated replication origins”. The same should be done for line 610.
25. Line 664. It should be noted in the figure legend that PCNA and RPA are included in the assay.
26. Line 670. Information on the 293T cells is missing from the “Cell lines and culture” section.
27. Line 702. Please add the concentration of puromycin used for selection.
28. Line 725 says sequencing was conducted using NovaSeq 6000, while line 729 says HiSeq 4000. Please modify the incorrect platform.
29. Line 732. Please provide a citation for the MAGeCK analysis (PMID: 25476604).

30. Line 761. Correct "bed" to "bead".
31. Line 788. Please provide details on how the images were analysed to determine % survival (colony count or stained area etc.).
32. Line 789. This statement is not true for the experiments in Figure S2B/S2C, where the data is normalised to an untreated control. Please update to reflect these too.
33. Line 911. The protocol states that the POLE proteins were diluted in an identical buffer and then concentrated again. Is this correct or should a component(s) of the diluting buffer be different?
34. Line 1039. "CTF18-RFC2_5-DSCC1-CTF8" should be "CTF18-RFC2_5-DSCC1-CTF8"
35. Methods section. Ensure all units are provided consistently across and within the different sections. Particularly temperature units (K vs °C), units of centrifugal force (g vs RCF vs RPM and the use of spaces before units of concentration).
36. Discussion section. Overall we recommend to reduce the repetition of results within the discussion section.

Version 1:

Reviewer comments:

Reviewer #2

(Remarks to the Author)

The authors largely addressed my concerns and I supported the acceptance of this study.

Reviewer #3

(Remarks to the Author)

Our major concerns have been satisfactorily addressed. We have only a few remaining comments regarding minor issues (listed below) that should be corrected before final approval. Once these minor issues are addressed, we consider the manuscript suitable for publication.

- Minor issue #21: A significance description was added to the legend of Figure 2E, as requested. However, it should also be removed from 2C where it is not applicable.
- Minor issue #34: This point does not appear to have been revised in the version we received.

Reviewer #1 (Remarks to the Author)

Summary: Using a combination of genetic, cell biology, biochemical and structural modeling approaches, the authors investigated DNA Polymerase ϵ regulation during leading strand synthesis. Their results focused on the role of the CHTF18-RFC complex and the two non-essential Pol ϵ accessory subunits, POLE3 and POLE4. The authors first mapped genetic factors required to support viability of POLE4 knockout cells but not wild type cells using a whole-genome CRISPR screen. Synthetic lethal interactions were observed between POLE3-POLE4 and iron metabolism genes, as well as the CHTF19-RFC2/5 complex. Effects on proliferation, cell cycle distribution and increased γ H2AX-positive cells were all observed in POLE3 and POLE4 KO cells treated with siNCOA4 (an essential iron regulator). The Iron Sulphur Cluster (ISC) in the catalytic subunit of Pol ϵ was demonstrated to be required for efficient DNA synthesis in vitro, despite being able to engage with its DNA substrate. In order to test the importance of iron regulation on Pol ϵ activity, the authors mutated two conserved cysteines in the Iron Sulfur Cluster of human Pol ϵ and showed that polymerase and exonuclease activities were abolished, despite the fact that the mutant enzyme can still bind DNA substrates.

Components of the CHTF19-RFC2/5 clamp loader were the other major class of genes identified in the genetic screen as required for viability in POLE3-POLE4 KO cells. The authors were unable to generate CHTF18-POLE3-POLE4 double KO cells, suggesting that both are critical for replication, and transient silencing of either POLE3-POLE4 in CHTF18 KO cells or silencing of CHTF18 in POLE3-POLE4 KO cells blocked proliferation and caused a reduction in replication fork extension and increased fork stalling.

The authors then used AlphaFold3 to model the interaction between POLE3-POLE4 and the catalytic POLE1 subunit of Pol ϵ to demonstrate and test, using crosslinking mass spectrometry and IP experiments, a mooring helix that is conserved in yeast. Using EMSAs, The POLE3-POLE4 subunits were shown to be critical for interaction with DS DNA at replication forks, and loss of these subunits resulted in reduced rates of primer extension and exonucleolytic activity.

The main conclusions made by the authors are that there are two modes of regulation of Pol ϵ processivity during leading strand replication. The first involves loading by CHTF18-RFC2/5. The second involves gripping of DS DNA at replication forks by the POLE3-POLE4 subunits of Pol ϵ . As accurate and efficient DNA synthesis and exonucleolytic proofreading by Pol ϵ are vital to prevention of diseases such as cancer, these results provide insight into the multiple layers of regulation of Pol ϵ activity and protein-protein interactions.

The questions being asked are compelling, and the results presented are novel and of interest to those studying DNA replication and how defects in this process contribute to human disease. The methodology used is appropriate to address the questions being asked and the conclusions are consistent with experimental results.

First of all, we would like to thank the reviewer for their enthusiastic support. We believe we have addressed their minor comments as follow

I have only minor suggestions where the results could be strengthened as follows:

1) Lines 126-130 – The statement ‘...pointing to the potential requirement for different replication-coupled DNA repair pathways in these cell lines...’ Can the authors please expand on this statement. I agree that performing this analysis in multiple cell lines was a good approach, and that the two similar cell lines used yielded similar but distinct results. Please expand on what is known regarding DNA repair pathway choice in each of these cell lines and how this may relate to the results presented.

We revised in accordance with the reviewer suggestion. we removed the indicated sentence and included some more speculative sentences in the results section.

2) Lines 174-176 – ‘... this data suggest that reduced stability of Pol ϵ might play at least a partial role in the observed synthetic lethality with iron metabolism factors’. This is a significant result and deserves more attention. The lower level of POLE1 is present in POLE3 and POLE4 KO cells even in cells not treated with the iron chelating agent, DFO, and the levels decrease further during the timecourse of DFO treatment. This reduction in POLE1 level was also shown in the POLE3 and POLE4 KO cells in Figure 4D. Although further investigation would be beyond the scope of this study, the possible mechanistic reasons for this POLE1 protein level should be discussed. Perhaps this could be explored using structural and/or modeling approaches.

We thank the reviewer for the comment. We had previously shown that loss of POLE3-POLE4 leads to destabilization of the full Pol Epsilon complex and reduction of its levels (Bellelli et al., Mol Cell 2018 May). More specifically, loss of POLE3-POLE4 reduces the stability of the full Pol ϵ complex and leads to its proteosomal degradation (Bellelli et al., Cell Reports, 2020). Loss of ISC assembly is known to destabilize ISC containing proteins so we believe the combination of these effects might lead to a further reduction of Pol ϵ levels. This might help to explain the synthetic lethality observed. We included a sentence in the discussion section to clarify this.

3) Figure 2C – Please alter the colors of the bars for S and G2/M as there is very little contrast between the two, making it difficult to appreciate the changes in % cells.

We changed the colours for S and G2/M as requested

4) Figure 6D – are the DNA substrates used in the EMSAs the same as those in Figure 3E?

Yes, the substrates are the same; we included a sentence in the text to clarify this.

5) Figure 7H and Supplementary Figure S6 – is it possible to quantify and plot the data for extension rate?

we quantified and plotted fork extension rate of 7H and previous figure S6 (now S8)

Reviewer #2 (Remarks to the Author):

In this study, Agnarelli et al aimed at understanding the function of POLE3 and POLE4, two accessory subunits of leading strand DNA polymerase, Pol epsilon, in DNA replication. First, the authors performed a CRISPR/Cas9 screen and identified genes that when depleted show synthetic lethal phenotypes with POLE4. They identified top candidate genes including those involved in iron metabolism such as NCOA4 and proteins involved in PCNA loading (CHTF18). They validated the effects of NCOA4 and CHTF18 depletion on cells. Furthermore, they mutated iron-sulfur cluster and found that mutations at this cluster affects DNA synthesis *in vitro*. Based on these results, they concluded that the dependence of POL4 KO cells on NCOA4 are likely due to the dependence of POLE1 on iron-sulfur cluster. It is known that POLE1 interacts with POLE3 and POLE4 through “mooring helix” domain. They mutated the binding interface between POLE1 and POLE4 and identified key residues that when mutated affect these interactions. Furthermore, based on Alpha-fold, they predicted POLE3-POLE4 binds dsDNA and mutations at the putative DNA binding surface affects DNA synthesis *in vitro*. Based on these results, they conclude that POLE3 and POLE4 are important to work with PCNA to stabilize POL epsilon at leading strand.

We thank the reviewer for their positive comments. We believe we have addressed their major and minor concerns as follows.

I have the following major concerns.
1) Fig. 3. NCOA4 has multiple functions. In order to connect synthetic lethality between NCOA4 and POLE4 to iron dependence, the effects of ISC mutant alone and in combination with POLE4 KO on DNA synthesis in cells must be tested.

We thank the reviewer for their comments. We believe the exact experiment suggested by the reviewer is not feasible. The iron sulfur cluster (ISC) mutant of POLE is completely dead *in vitro* and, as such, it would not be compatible with cell viability. However, to further corroborate an iron dependent mechanism at the basis of the observed synthetic lethality we performed DNA fiber experiments in cells subjected to silencing of NCOA4 and treated or not with DFO, an iron chelator. Strikingly silencing of NCOA4 led to a reduction of fork extension rates in POLE3-POLE4 KO but not wt cells. Importantly, a similar result was obtained by using deferoxamine (DFO), pointing to an iron-dependent mechanism (Fig S2A and S2E). Finally, we would like to point

out that in addition to NCOA4, multiple iron metabolism genes were identified in the crispr screen, thus suggesting a selective requirement of POLE3-POLE4 KO cells for controlled iron homeostasis.

2) Fig. 4. Because CHTF18 and POLE3 and POLE4 function in other processes in addition to DNA replication, the effects of CHTF18 mutants that cannot bind to PCNA on replication forks and cell viability must be tested alone or in combination with POLE3 and POL4 KO.

We thank the reviewer for the suggestion. In accordance with structural work from the Li lab <https://www.pnas.org/doi/epdf/10.1073/pnas.2319727121> we introduced the following mutations in CHTF18: A415K, M419D, V422E (residues highlighted in red in the attached figure) to abrogate interaction of CHTF18 with PCNA

We then tried to generate cell lines expressing CHTF18 WT and the mutant under basal and dox-regulated expression. However we observed no expression of the mutant in both cases, suggesting instability of the mutant protein in cells (please see attached wb). Thus, unfortunately, despite multiple attempts, we could not performed this experiment.

3) Fig. 6. What are the effects of PolE1 W1271R and POLE4 D53K mutation on DNA synthesis in vitro? How do these mutants affect DNA replication forks in cells? Are these mutants also show synthetic growth defects with CHTF18 depletion?

We thank the reviewer for the comment. Expression of POLE1 W1271R and POLE4 D53K in vitro would lead to the purification of a POLE3-POLE4 lacking complex as interaction in cells is completely lost. To answer the reviewer question, we have generated POLE4 KO cells expressing POLE4 WT and a D53K mutant. we now show that the POLE4 D53K mutant is not able to rescue the synthetic lethality with depletion of CHTF18 and lead to replication fork defects (revised Fig. S5A and S5B).

4) Fig. 7. Does POLE4 DB mutant affect histone binding? Does this mutant show synthetic defects with CHTF18 depletion?

We thank the reviewer for the suggestion. we now show that a pol epsilon complex with POLE4 DBM is still able to interact with histone H3 (Figure S7). We have also expressed the mutant in POLE4 KO cells and show that the synthetic lethality with CHTF18 depletion is conserved (Figure S5A-B).

5) Fig. 7H. Please clarify whether PCNA was included in vitro reaction. If not included, the experiments should be repeated by including PCNA in the reaction.

We apologize for the mistake. PCNA and RPA were included in all the reactions. This is now highlighted in the figure legends. To further point to a PCNA dependent process we performed kinetics experiments in the presence and absence of CHTF18 and PCNA alone. As show in figure S8 addition of CHTF18 without PCNA does not stimulate DNA synthesis by Pol ϵ . Similarly addition of PCNA in the absence of its physiological loader CHTF18-RFC does not promote leading strand extension by Pol ϵ .

Minor concerns.

1)Cite the reference PMID: 30115745 for the role of POLE3 and POLE4 in parental

histone transfer. What is evidence of reference 5 showing that POLE3 and POLE4 in parental histone transfer?

We apologize with the reviewer for the omission. We have now included the reference as requested. Reference 5 is showing that POLE3-POLE4 can act as histone chaperone in vitro.

2) "...survival of POLE3-POLE4 KO cells and, by extension, Pol ϵ function". POLE3 and POLE4 have other functions. Therefore, the statement above is not accurate.

We apologize with the reviewer for the mistake. We have corrected the statement as requested.

Reviewer #3 (Remarks to the Author)

In this manuscript, Agnarelli et al aim to identify genetic factors involved in leading strand DNA synthesis by DNA polymerase epsilon (Pol-epsilon) in human cells. By performing CRISPR screens in POLE4 knockout cell lines, the authors uncover that both iron metabolism dysfunction and loss of the alternative clamp loader complex CHTF18-RFC sensitise cells. Building upon these findings, the authors assess whether the identified factors are functionally linked to Pol-epsilon by examining their genetic interactions with POLE3-POLE4 and through structural analysis of key residues. Regarding iron metabolism, the authors demonstrate that the combination of iron homeostasis dysregulation and POLE3/POLE4 knockout leads to severe growth defects. With respect to CHTF18-RFC, although its association with the replisome and its role in efficient replication fork progression have been previously reported (Fujisawa et al., 2017, NAR, PMID: 28199690; Baris et al., 2022), the structural and biochemical analyses presented in this study suggest a complementary relationship between CHTF18-RFC and the POLE3-POLE4 complex. Overall, I believe this study provides informative and reasonable evidence that these factors contribute to efficient Pol-epsilon-mediated DNA synthesis. The findings offer valuable insights that merit consideration for publication in Nature Communications. However, in our opinion, the following issues should be addressed prior to publication:

We would like to thanks the reviewer for their enthusiastic support. We believe we have addressed their major and minor comments in the revised version of our manuscript.

Major issues

1. The authors demonstrate a genetic interaction between iron levels and POLE3-POLE4, resulting in both proliferation defects and elevated γ H2AX levels in G2 phase, which may stem from inefficient replication during S phase. However, I find the evidence linking iron metabolism specifically to leading-strand DNA synthesis to

be somewhat indirect. I suggest that a more direct assessment, such as a DNA fiber assay in POLE3/POLE4 knockout cells with and without siNCOA4, could help clarify whether iron metabolism directly impacts leading-strand synthesis.

we thank the reviewer for their comment. As requested we have now performed DNA fiber experiments in WT and POLE3-POLE4 KO cells upon silencing of NCOA4. Consistently with defective leading strand synthesis in this condition, silencing of NCOA4 led to a dramatic reduction of fork extension rates in POLE3-POLE4 KO cells but not in the wild-type counterparts. This is in agreement with our FACS data and with a critical role for iron homeostasis in promoting leading strand synthesis in POLE3-POLE4 KO cells.

2. To investigate the potential impact of disrupted iron homeostasis on Pol-epsilon, the authors introduce mutations in amino acids that constitute the iron-sulfur cluster motif of the purified protein. However, it remains unclear whether the observed effects of these mutations are directly attributable to the loss of iron within the protein structure. It is also conceivable that individual mutations may induce broader structural alterations independent of iron coordination. I would encourage the authors to consider an experimental approach that can distinguish between these possibilities.

We thank the reviewer for the comment. The domain containing the Fe-S cluster mutant used in the manuscript must be correctly folded since we have shown in the EMSA experiments that the mutant complex is able to bind ssDNA, dsDNA and replication fork structures in a manner that mimics the wild type. Therefore, we believe the mutation does not cause broader structural alterations in the POLE1 subunit.

Despite being able to bind to DNA, the Fe-S mutant is unable to extend a primed DNA substrate as shown in the extension assays. As we briefly speculate in the discussion section it is possible that loss of the iron sulphur cluster could prevent the movement of the 'finger' domain that has the Fe-S at its 'knuckle' acting as a pivot. In order to properly assess this model though would require determining multiple experimental Cryo-EM structures of the Fe-S mutant bound to its DNA substrate, and/or the development of a complicated reporter assay for finger movement such as the introduction of FRET pairs into the complex. While we thank the reviewer for this interesting comment, and the discussion it has stimulated in our group, we believe this work is well beyond the scope of the present study and represents an entire additional study to be conducted at a future date. We hope this sufficiently addresses the reviewer criticism at this point.

3. The authors infer that that role of CHTF18 in stimulating efficient synthesis involves

its function in PCNA loading. However, evidence supporting this claim is insufficient. This statement could be supported via in vitro experiments showing the dependence of efficient synthesis on PCNA (e.g. by conducting the experiment in Figure 7H +/- PCNA or with PIP-box mutated POLE).

To address the reviewer comment we have performed extension assays with full pol Epsilon in the presence and absence of PCNA and CHTF18 alone. As shown in figure S8 both CTF18 and PCNA alone were not able to stimulate DNA synthesis by full Pol Epsilon.

Minor issues

1. Figure 2A/2B. Knockdown of NCOA4 is less efficient in WT compared to POLE3 and POLE4 KO cells. Could residual protein in these cells mean that proliferation is not affected to the same extent?

We believe this does not affect the validity of our experiments as silencing is profound in all conditions. furthermore we would like to point out that siRNA experiments are confirmative of a genome wide unbiased crispr screening that identified NCOA4 as synthetic lethal in POLE4 ko but not wt cells.

2. Figure 2G. An untreated control should be provided in this panel, to highlight the differences induced by iron chelation. The data should also be plotted as “change in % confluency” for consistency with the associated legend (line 555) and the other figures.

We revised the panel in accordance to reviewer suggestion.

3. Figure 4A,B,C,D,H,I. Figures are labelled with CTF18, not CHTF18.

We apologise for the labelling mistake. We revised accordingly.

4. Figure 7H and Figure S6. Figure is labelled with CTHF18, not CHTF18.

We apologise for the labelling mistake. We revised accordingly

5. Line 69. Correct the spelling of “independent”.

We corrected the spelling mistake in line 68

6. Line 69. The wording here is somewhat unclear, without the context of the previous

paper. The phrasing can be simplified. Perhaps “these data are independent of Pol-epsilon expression levels and suggest functions for POLE3-POLE4 in controlling DNA replication and genomic stability”.

We revised accordingly to reviewer suggestion

7. Lines 101-108. The introductory lines of the first section largely repeat information from the introduction. These could therefore be shortened.

We thank the reviewer for the comment. We agree this is repetitive. As such we have removed the whole section and started directly with the crispr screen.

8. Line 165. The authors state that redundancy and different expression levels of IRP1 and IRP2 reduces the deleterious effects of depleting IRP2 in RPE1 cells, but no evidence is provided for either redundancy or different expression. If this is speculation, I suggest removing “the” from the sentence.

We remove “the” as suggested by the reviewer since indeed this is a speculation

9. Line 225. These results, combined with the CRISPR screen, do indeed point to a synthetic lethal interaction between POLE3-POLE4 and CHTF18. However, this section would benefit from quantitative details of the CRISPR editing attempted, e.g. how many times the experiment was conducted and how many clones were screened (if they could be isolated).

We attempted once with POLE3 KO and once with POLE4 KO cells and screened ~30 clones for genotype. All isolated clones expressed CHTF18 close to wt levels. We added a sentence to further clarify this

10. Line 226. The sentence regarding loss of CHTF18 appears to contradict itself by first implying that there is no obvious defects in cell growth but then explaining that the data show a growth defect. This sentence should be refined for clarity of the situation (i.e. that when kept under normal conditions growth appears fine, but defects occur when seeded at low confluency).

We revised accordingly

11. Line 243. The text directs to “Fig S2”, but it should direct to “Fig. S3”.

We revised accordingly

12. Line 251. The text states that CHTF18 KO cells present a significant reduction in fork speed, but the significance of this comparison is not displayed in Figure 4H.

Please add this comparison to the figure. Also add the same comparison to Figure 4I, which is described as not changing on line 252.

We apologize with the reviewer. We have included now statistical analyses of these comparisons. we noticed a slight but significant reduction of fork symmetry also in basal conditions in CHTF18 KO cells. this is now highlighted in the text as well.

13. Line 276. Please edit this sentence to be more clear about where the revealed cross-links are between. Perhaps "...revealed cross-links between the linker region near the putative mooring helix and both POLE3 and POLE4" is more clear.

We revised accordingly

14. Line 283. It should be noted that a different cell line is being used for the IP experiments compared to the previous experiments. While we assume that the fundamentals of POLE1 are the same between the cell lines, it is important to make the difference clear in the main text.

We included in the text that interaction was tested in 293T cell extracts

15. Line 289. The text states that mutating R1286 in POLE1 affects the intereaction with POLE3, but this doesn't seem supported by the IP shown in Figure 5D, where seemingly comparable amounts of POLE3 are pulled down by WT and R1286A. Please either alter the text to reflect this or provide the quantification etc that was used to determine the detrimental effect.

We revised the text accordingly. Thus we included only the E77 mutant that showed a clear reduction in the interaction.

16. Line 345. Please cite the "previous work".

We have now included the reference in the text

17. Line 349. Please cite the "recently published work".

We have now included the reference in the text

18. Line 373. Again, if a difference in dependencies or processes is only speculative, this should be more clear. Here "the" could be changed to "a" to better reflect this.

We revised accordingly

19. Line 400. Reference 20 is cited, but it should be 19.

We revised accordingly

20. Lines 536-542. Since Figure A and B present the same experiment in the same way, the description of B can be shortened to reduce repetition. E.g. “As A, with RPE1...”.

We revised accordingly

21. Line 544. A description of the significance labels is given, but Figure 2C does not have such labels. Perhaps this text relates to Figure 2E, for which a description of the labels is missing (Line 547). Please fix as appropriate.

We apologize for the mistake. We revised accordingly

22. Line 571. This sentence implies that the substrate and dNTPs are used at equimolar concentrations, rather than the enzyme and substrate. Please reword for clarity. This also appears on line 632 and line 656.

We revised accordingly

23. Line 601. Please add an explanation of the significance labels for Figure 4E.

We included significance labels in the revised manuscript

24. Line 605. To enhance clarity of the experiment conducted, please replace “replication forks at newly activated replication origins” with “replication forks either side of newly activated replication origins”. The same should be done for line 610.

We revised accordingly

25. Line 664. It should be noted in the figure legend that PCNA and RPA are included in the assay.

We included a sentence to clarify this

26. Line 670. Information on the 293T cells is missing from the “Cell lines and culture” section.

We included the information in the revised version

27. Line 702. Please add the concentration of puromycin used for selection.

We included the information in the revised version

28. Line 725 says sequencing was conducted using NovaSeq 6000, while line 729 says HiSeq 4000. Please modify the incorrect platform.

We apologize for the mistake. Sequencing was performed using NovaSeq 6000. We revised the text accordingly.

29. Line 732. Please provide a citation for the MAGeCK analysis (PMID: 25476604).

We included the reference as requested.

30. Line 761. Correct “bed” to “bead”.

We corrected the mistake

31. Line 788. Please provide details on how the images were analysed to determine % survival (colony count or stained area etc.).

We clarified in the revised manuscript

32. Line 789. This statement is not true for the experiments in Figure S2B/S2C, where the data is normalised to an untreated control. Please update to reflect these too.

We revised accordingly

33. Line 911. The protocol states that the POLE proteins were diluted in an identical buffer and then concentrated again. Is this correct or should a component(s) of the diluting buffer be different?

We apologize for the mistake. We revised accordingly

34. Line 1039. “CTF18-RFC2_5-DSCC1-CTF8” should be “CHTF18-RFC2_5-DSCC1-CHTF8”

We revised accordingly

35. Methods section. Ensure all units are provided consistently across and within the different sections. Particularly temperature units (K vs °C), units of centrifugal force (g vs RCF vs RPM and the use of spaces before units of concentration).

We revised accordingly

36. Discussion section. Overall we recommend to reduce the repetition of results within the discussion section.

we thank the reviewer for the suggestion. we revised where possible the discussion section.